# Structure of the Nipah virus polymerase complex

Esra Balıkçı [ID][1,4], Franziska Günl [ID][2,4], Loïc Carrique[1], Jeremy R Keown [ID][3], Ervin Fodor [ID][2] &
Jonathan M Grimes [ID][1✉]

## Abstract

**Nipah virus is a highly virulent zoonotic paramyxovirus causing severe respiratory and neurological disease. Despite its lethality, there is no approved treatment for Nipah virus infection. The viral polymerase complex, composed of the polymerase (L) and phosphoprotein (P), replicates and transcribes the viral RNA genome. Here, we describe structures of the Nipah virus L-P polymerase complex and the L-protein's Connecting Domain (CD). The cryo-electron microscopy L-P complex structure reveals the organization of the RNA-dependent RNA polymerase (RdRp) and polyribonucleotidyl transferase (PRNTase) domains of the L-protein, and shows how the P-protein, which forms a tetramer, interacts with the RdRp-domain of the L-protein. The crystal structure of the CD-domain alone reveals binding of three Mg ions. Modelling of this domain onto an AlphaFold 3 model of an RNA-L-P complex suggests a catalytic role for one Mg ion in mRNA capping. These findings offer insights into the structural details of the L-P polymerase complex and the molecular interactions between L-protein and P-protein, shedding light on the mechanisms of the replication machinery. This work will underpin efforts to develop antiviral drugs that target the polymerase complex of Nipah virus.**

**Keywords** Nipah Virus; Polymerase Structure; Cryo-EM
**Subject Categories** DNA Replication, Recombination & Repair; Microbiology, Virology & Host Pathogen Interaction; Structural Biology

## Introduction

Nipah virus (NiV) infections cause atypical pneumonia and severe febrile encephalitis, with a mortality rate of up to 75% over the past several decades (Chua et al, 2000; Eaton et al, 2006). NiV was first identified during an outbreak affecting pigs and pig farmers in Malaysia and Singapore in 1998–1999 (Chua et al, 2001). During this outbreak, 265 cases of encephalitis occurred, resulting in the deaths of 105 people and the culling of nearly one million domestic pigs (Ang et al, 2018; Chua et al, 2000). Later, sporadic outbreaks of NiV occurred in India, including the last one in Kerala in 2023, while Bangladesh has been hit with outbreaks almost every year since 2001 (Cui et al, 2024). Being a recurring threat with high

mortality rates and a lack of specific treatments, as well as a zoonosis with the potential of human-to-human transmission, Nipah virus represents a significant public health problem.

NiV and the closely related Hendra virus (HeV) are members of the genus Henipavirus within the *Paramyxoviridae* family. Henipaviruses contain a single-stranded, negative-sense non-segmented RNA (nsNSV) genome that encodes six structural proteins. The RNA genome is encapsidated by nucleoprotein (N-protein), forming a helical nucleocapsid (Ker et al, 2021). The RNA genome is both transcribed into viral mRNAs and replicated by the viral RNA polymerase complex comprising the Large protein (L-protein) and phosphoprotein (P-protein). The L-protein, a pivotal component of the polymerase complex, contains three catalytic domains, the RNA-dependent RNA polymerase (RdRp-domain), the polyribonucleotidyl transferase (PRNTase-domain), and methyltransferase (MTase-domain), in addition to two structural domains, the connecting domain (CD-domain) and C-terminal domain (CTD-domain) (Fig. 1A). During transcription, the L-protein catalyses the synthesis of a 5′ capped and 3′ polyadenylated mRNA. Viral RNA replication is a two-step process with the initial synthesis of a full-length copy of the genome, known as the antigenomic RNA of positive polarity. This antigenomic RNA, in turn, acts as a template for the subsequent synthesis of the genomic RNA (Jordan et al, 2018). These processes rely on the presence of the non-catalytic P-protein, which plays a pivotal role as a coordinating hub for the L-protein, the nucleocapsid and RNA-free N-protein (Yabukarski et al, 2014). The P-protein is composed of an ordered N-terminal peptide, separated by an intrinsically disordered region from a central oligomerization domain (OD-domain) and a C-terminal X domain (XD-domain) (Bruhn et al, 2014; Bruhn et al, 2019). The N-terminus of the P-protein acts as a chaperone by binding to RNA-free N-protomers (N0), which subsequently encapsidate the newly synthesized antigenomic and genomic viral RNA (Bloyet, 2021; Yabukarski et al, 2014). The C-terminal XD-domain has been shown to be important in mediating the interaction between the RNA bound nucleocapsid and the L-protein (Bourhis et al, 2022; Wolf and Plemper, 2024). Thus, the P-protein functions as a central hub in coordinating the assembly of ribonucleoprotein complexes (Ogino and Green, 2019).

Recent years have seen significant advancements in understanding the replication and transcription mechanisms of nsNSVs, with the publication of the structures of L-P complexes from a number of mononegaviruses. These include vesicular stomatitis virus (VSIV) (Jenni et al, 2020; Liang et al, 2015), rabies virus (RABV) (Horwitz et al, 2020), human respiratory syncytial virus (HRSV), (Cao et al, 2024; Gilman et al, 2019; Pan et al, 2020), human metapneumovirus

[1]Division of Structural Biology, University of Oxford, Oxford, UK. [2]Sir William Dunn School of Pathology, University of Oxford, Oxford, UK. [3]School of Life Sciences, University of Warwick, Coventry, UK. [4]These authors contributed equally: Esra Balıkçı, Franziska Günl. ✉E-mail: jonathan.grimes@strubi.ox.ac.uk

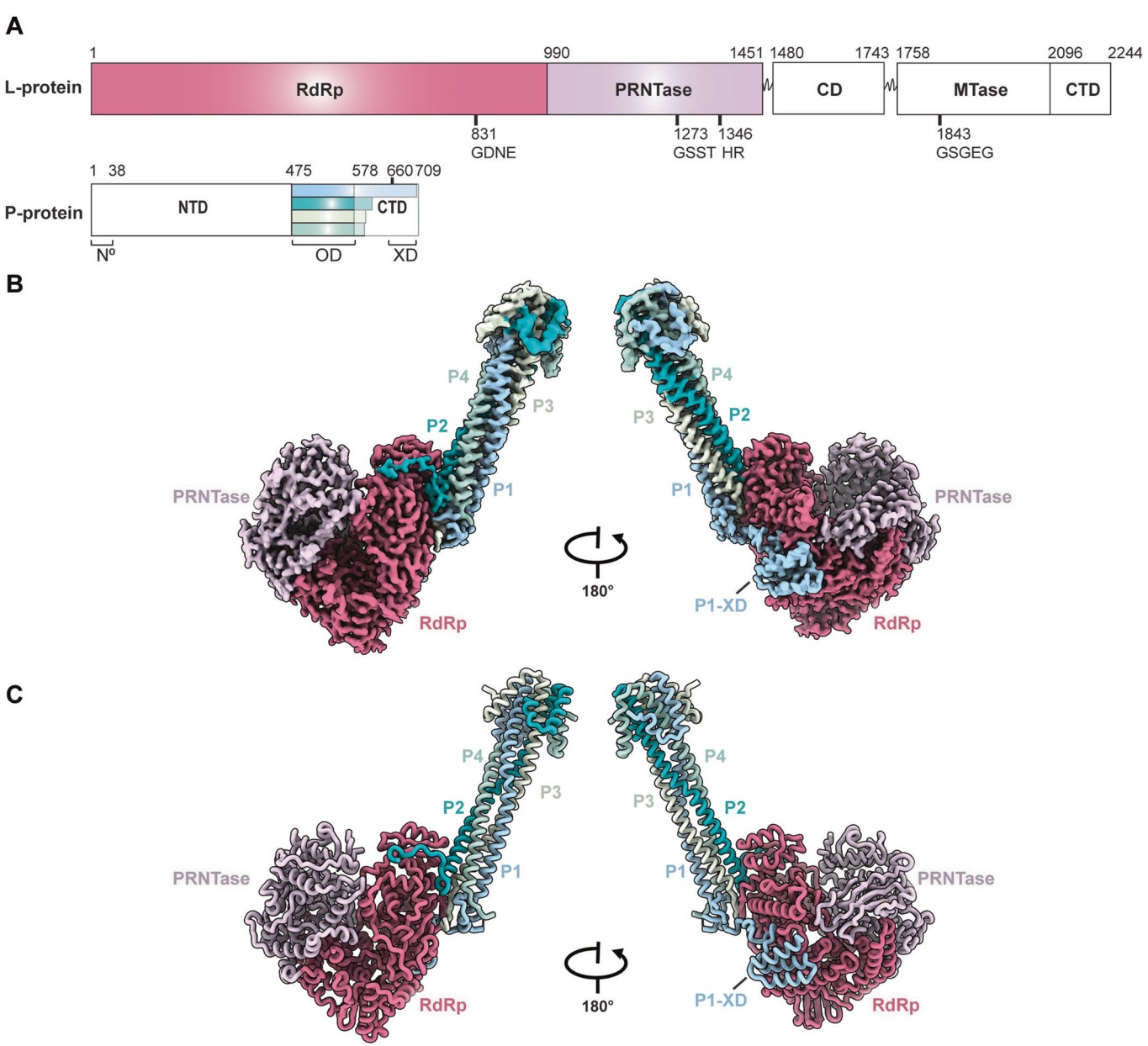

**Figure 1. Structure of the NiV L-P complex.**

(A) Schematic representation of the domain organization of the NiV L- and P-proteins. The domains of L-protein and the four copies of P-protein resolved in the L-P structure are shown in colour. (B) Cryo-EM density map of the L-P complex. (C) Cartoon representation of the modelled structure; the domains are colour coded as in (A).

(HMPV) (Pan et al, 2020), parainfluenza virus 3 and 5 (Abdella et al, 2020; Xie et al, 2024), Newcastle disease virus (NDV) (Cong et al, 2023), Mumps virus (MuV) (Li et al, 2024) as well as Ebola virus (EBOV) (Peng et al, 2023; Yuan et al, 2022). These studies reveal the conserved RdRp- and PRNTase-domains, the flexibility of the CD-domain, MTase-domain and CTD-domain as well as conformational changes when bound to their RNA templates (Te Velthuis et al, 2021). Here, we present the molecular structure of the NiV L-P complex imaged by cryo-electron microscopy (cryo-EM) and the crystal structure of the CD-domain of the L-protein. These results reveal the structural arrangement of the RdRp- and PRNTase-domains of the polymerase complex, and how the tetrameric P-protein binds to the L-

protein. By modelling the CD-domain onto an Alphafold 3 model of an RNA-L-P complex, our data suggest insights into the early steps of capping and the conformational changes needed for the elongation of the synthesised viral RNA.

## Results

### Overall structure of the NiV L-P complex

To determine the structure of the NiV polymerase complex, we co-expressed the L- and P-proteins from the Malaysian strain of NiV

in Spodoptera frugiperda 9 (Sf9) cells. Purification of the polymerase complex yielded a stoichiometric preparation suitable for structural and biochemical studies (Fig. EV1A,B). To confirm that the L-P complex was functional, we performed a template-dependent in vitro polymerase assay using a 12 nucleotide 3′ leader sequence of the NiV genome as a template (Fig. EV1C). The activity assays revealed that the L-P complex produced RNA products of varying lengths in agreement with earlier studies (Jordan et al, 2018). Substituting the catalytic active site aspartic acid with alanine at position 832 (D832A), resulted in no RNA product, indicating a loss of polymerase activity (Fig. EV1C). Substituting adenosine triphosphate with remdesivir triphosphate did not terminate strand elongation but resulted in a similar pattern of RNA products with slightly increased mobility (Fig. EV1D).

We determined the structure of the L-P complex at a 2.5 Å resolution using single-particle cryo-EM. The final reconstruction (Fig. EV2) enabled us to build a model for a portion of the L-protein bound to an asymmetric tetramer of P-protein (Fig. 1A–C). The N-terminal RdRp-domain and PRNTase-domains of the L-protein were almost entirely resolved with the PRNTase-domain bound to two zinc ions. The CD-domain, MTase-domain, and the CTD-domain were not visible in our map, despite being present in our construct. The cryo-EM map revealed no density for the N-terminal 475 residues of P-protein, possibly due to the intrinsic disorder of this region. However, we could build models for four separate monomers of the P-protein, varying in length, and starting from the beginning of the OD-domain. The longest visible P-protein monomer spans amino acid residues 477–707; of these, 477–510 correspond to short helical bundles located at the N-terminal end of the OD-domain, 510–575 form the OD-domain, while 660–706 represent the XD-domain of the P-protein (Fig. 1).

Analysis of the L-P complex by SDS-PAGE after size exclusion chromatography shows that both L- and P-proteins have maintained their integrity suggesting that the missing domains reflect the highly dynamic nature of the L-P complex (Figs. 1 and EV1A,B). This flexibility has been observed previously with L-P complexes of other nsNSVs (Fig. EV3).

## Structural details of the polymerase complex

The structural analysis revealed the organization of conserved motifs and residues critical for the catalytic function and the regulatory mechanisms of the L-protein. The RdRp-domain of the L-protein folds into the right-handed "fingers-palm-thumb" architecture found in many RNA virus polymerases (Figs. 2A and EV4A) and contains seven specific structural motifs (A-G) (Fig. 2B; Appendix Fig. S1). Motif A is involved in binding divalent cations via the conserved aspartate residues (Asp722 in NiV), which is essential for the catalytic activity of the RdRp-domain (Arnold et al, 1999; Gohara et al, 2000). Motif B binds template RNA and incoming nucleoside triphosphates (NTPs), and together with motifs A, D and E, it contributes to the structural integrity and the selection (Campagnola et al, 2015; Ferrer-Orta et al, 2004). The active site residues ($^{831}$GDNE$^{834}$) are found at the tip of a β-hairpin formed by motif C, buttressed by structural elements from the fingers subdomain and the N-terminal domain (NTD-domain) of the RdRp-domain that form the tunnel that allows the template RNA to move towards the catalytic site. Motifs F and G, bridging

the palm and fingers regions, contribute to the RdRp-domain's structural stability and flexibility, facilitating effective positioning of template RNA and NTPs (Bruenn, 2003).

The structural elements, including the priming loop (residues 1256-1290), which is proposed to support the initiating nucleotide (Cressey et al, 2022; Ogino and Green, 2019), and the intrusion loop (residues 1337-1362) of the PRNTase-domain, which contains the HR motif that forms a covalent bond with the RNA (Ogino and Green, 2019), as well as the supporting helix (residues 588-600) of the RdRp-domain (Fig. EV4B), which provides structural support and stability to the active site (Liang, 2020), lacked sufficient density to be fully modelled in our L-P complex structure (Appendix Fig. S2A,B). Structures of previously solved L-P polymerase complexes from nsNSVs reveal two possible conformational states of the catalytic chamber. In the initiation state, the priming loop and the supporting helix largely occupy the product RNA binding groove (Horwitz et al, 2020; Jenni et al, 2020; Liang et al, 2015). After reaching the elongation state, the flexible supporting helix becomes disordered, while the priming loop is retracted and repositioned towards the PRNTase-domain to accommodate the nascent double-stranded RNA duplex formed between the template and nascent product (Gilman et al, 2019; Pan et al, 2020; Yuan et al, 2022). The absence of sufficient density for these structural elements in the NiV L-P map suggests a degree of flexibility, which is consistent with the absence of bound RNA template and NTP. AlphaFold 3 prediction suggests that the missing residues of these structural elements would indeed be stabilized in the presence of a bound template RNA, which is consistent with a composite model based on the EBOV RNA bound L-protein structure (Peng et al, 2023) (Appendix Fig. S2C,D). Interestingly, the RdRp-domain of the NiV harbours a 105 amino acids (605-710) insertion within the palm subdomain, a feature unique to Henipaviruses (Appendix Fig. S3A). This region is disordered in our maps, indicating its inherent flexibility. Deletion of this sequence resulted in the inhibition of RNA replication in a cell-based minireplicon assay, suggesting its importance for polymerase activity (Appendix Fig. S3B).

The entry and exit channels for template RNA are located adjacent to the fingers and thumb sub-domains of the RdRp-domain. As the nascent RNA extends, it passes through a positively charged tunnel towards the product exit channel, where the MTase-domain and CTD-domain are located (Fig. 2C,D). The MTase-domain and CTD-domain exhibit dynamic behaviour and can adopt multiple conformations where the CD-domain acts as a hinge between the PRNTase-domain and the MTase-domain and CTD-domain portion. Despite the lack of observable density for the C-terminal domains in our cryo-EM map, we determined the structure of the CD-domain by X-ray crystallography at a resolution of 1.85 Å (Fig. 3A,B). Structural analysis revealed that CD-domain adopts a fold similar to the structures observed within the Paramyxovirus and Rhabdovirus families (Fig. 3C,D), despite displaying minimal sequence conservation (Appendix Fig. S1). Interestingly, we observed 3 structural magnesium ions (designated as Mg1, Mg2, and Mg3), with octahedral coordination characteristic of divalent cations. In particular, Mg1 ion is bound to the N-terminal α2 helix coordinated by the side chains of Asn1526 and Ser1529, as well as the main chain carbonyl oxygen of Ser1523, along with two water molecules (Fig. 3B). Refining these sites as manganese ions resulted in inflated B-factors and distorted

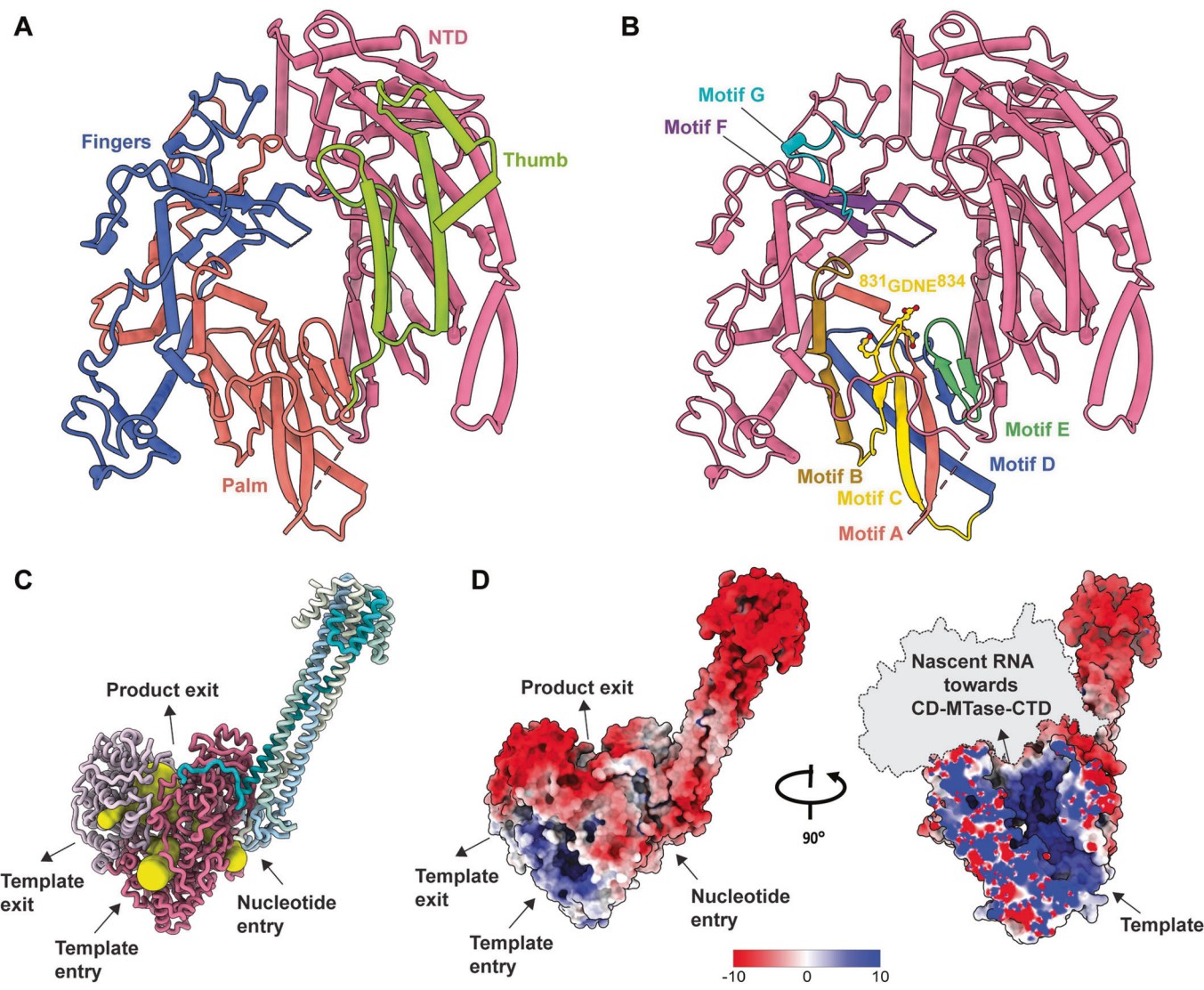

**Figure 2. Structural organization of the NiV L-P complex.**

(A) The sub-domains of the RdRp-domain of L-protein are shown in different colours. (B) The catalytic motifs of the RdRp-domain A-G are highlighted in different colours. The amino acid residues of the $^{831}$GDNE$^{834}$ sequence are shown as sticks. (C) Nucleotide entry, template entry, template exit, and product exit channels are indicated by yellow tunnels. (D) The electrostatic iso-surface representation shows that the L-P complex is highly negatively charged, while the RNA binding pocket of the polymerase is highly positively charged (blue: positive, red: negative). On the right is the path of the nascent RNA leading towards the CD-, MTase- and CTD-domains within the positively charged groove.

geometry. This part of the CD-domain is known to be dynamic from the structures of other nsNSV CD-domains (Fig. 3D) and this flexibility is consistent with its potential role in orchestrating conformational alterations of the C-terminal domains, essential for coordinating the diverse stages of RNA synthesis.

## Molecular basis for P-protein binding

The structure shows that P-protein exhibits a high degree of structural flexibility in which each monomer can adopt different conformations when interacting with different regions of the RdRp-domain of the L-protein (Fig. 4A; Appendix Fig. S4). All four monomers of the P-protein contain an OD-domain from residues

510 to 575, which form a tetrameric four-helix bundle, as shown previously (Bruhn et al, 2014). However, the N-terminal and C-terminal regions outside the OD-domain have not been structurally well characterized due to their intrinsic flexibility. Our structure reveals that one of the monomers of the P-protein (P1) has a complete ordered XD-domain that interacts with L-protein through a series of hydrophobic and electrostatic interactions (Fig. 4B). P1 forms a tentacle-like structure consisting of three consecutive α-helices that surrounds the channel through which NTPs access the active site. Arg669 and Asn702 of P1 form salt bridges with Asp339, and Arg308 of RdRp-domain of the L-protein, respectively (Fig. 4B), while closer towards the OD-domain Arg600 of P1 mediates salt bridges with Glu760 and Glu733 of

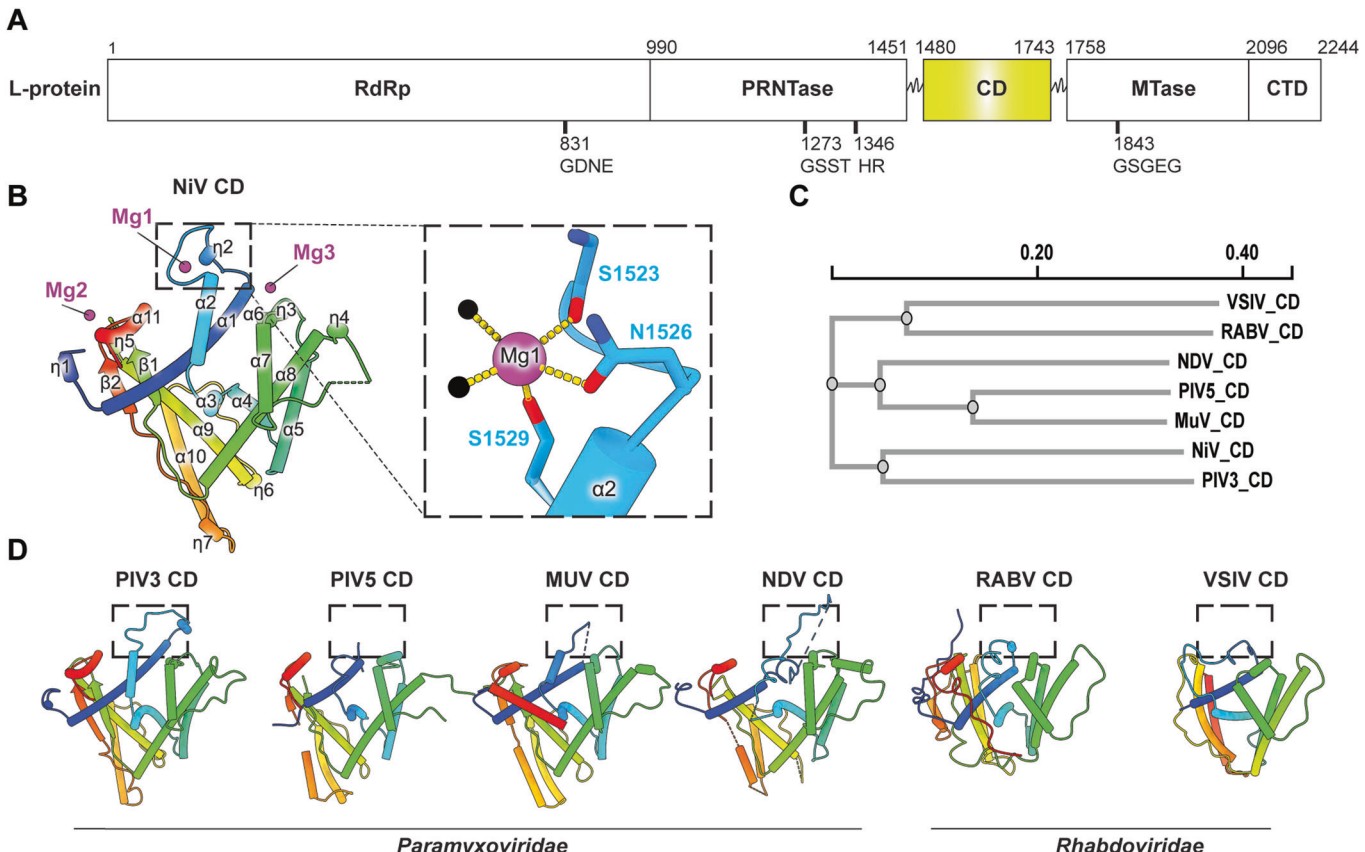

**Figure 3. Structure of NiV CD-domain of the L-protein.**

(A) Schematic representation of domain boundaries of NiV CD-domain of the L-protein. (B) The cartoon representation of the NiV CD-domain is shown in rainbow colours from the N- to C-termini, with secondary structures such as helices, β-sheets and 3₁₀ helices labelled as α, β and η, respectively. The three Mg ions (Mg1, Mg2 and Mg3) bound to the NiV CD-domain are illustrated as magenta spheres. A close-up view of the Mg1 bound to the α2 helix is shown, highlighting the residues coordinating the Mg1, which are depicted as sticks. Water molecules are shown as black spheres. (C) Sequence-based phylogram of the structurally available CD-domains generated in Clustal Omega (Sievers et al, 2011). (D) Structural comparisons to the CD-domains of nsNSV L- proteins including members of the *Paramyxoviridae* (PIV3: PDB 8KDC, PIV5: PIDB 6V85, MUV: PDB 8IZL, NDV: PDB 7YOU) and *Rhabdoviridae* (VSIV: PDB 5A22, RABV: PDB 6UEB) families. CD-domains exhibit structural flexibility of the α2 helix in their N-terminal region, as indicated by the dashed box.

RdRp-domain (Fig. 4C). In addition, hydrogen bonds are formed between Pro640, Ala649, Thr670, His671 of P1 and Arg867, Ala879, Asn346, Leu300 of RdRp-domain, respectively (Fig. 4B,C). Residues 575–578 of P3 form a β-strand which is sandwiched between a parallel β-strand from P1 (597-599) and an anti-parallel strand from RdRp-domain (residues 385–388) to form a three-stranded β-sheet. These interactions are mediated by hydrogen bonds and hydrophobic interactions (Fig. 4D). Residues 570–595 of P2 bind on the opposite side of the RdRp-domain compared to P1. The His570 of P2 mediates a salt bridge with Glu448 and is also involved in a hydrogen bond with Tyr389 of the RdRp-domain. The residues Asn590 of P2 interact with the main chain of Met459 of the RdRp-domain, via a hydrogen bond, while Leu594 and Pro579 of P2 mediate hydrophobic interactions with Leu525 and Tyr732 of the RdRp-domain, respectively (Fig. 4E).

We next investigated the functional importance of these residues by generating a panel of 18 P-protein variants, in which these residues were mutated to alanine or glycine. These variants were tested using a Split-Luciferase complementation assay and a minigenome replication assay (Appendix Fig. S5A). Most variants,

except H671A and S565A, exhibited reduced Gaussia luciferase complementation relative to the wild-type protein. Among these, the A649G, P579A, I578A, and H570A variants completely abolished polymerase activity in the minigenome assay, while H671A caused a significant reduction in activity (Appendix Fig. S5B). These findings collectively indicate that the multifaceted interactions of P-protein are essential for its role in viral replication.

## A model for RNA binding and processing

To gain further insights into the mechanism of RNA synthesis, we constructed a composite model by integrating the cryo-EM structure of RdRp-domain and PRNTase-domain, and the crystal structure of CD-domain, onto an AlphaFold 3 (AF3) model of the full NiV L-protein (Appendix Figs. S6 and S7) where the MTase-domain and CTD-domain are predicted to be ordered (Fig. 5). This full-length AF3 L-protein model predicts the localization of disordered structural elements—specifically, the intrusion loop, priming loop, and insertion sequence—on the cryo-EM structure of

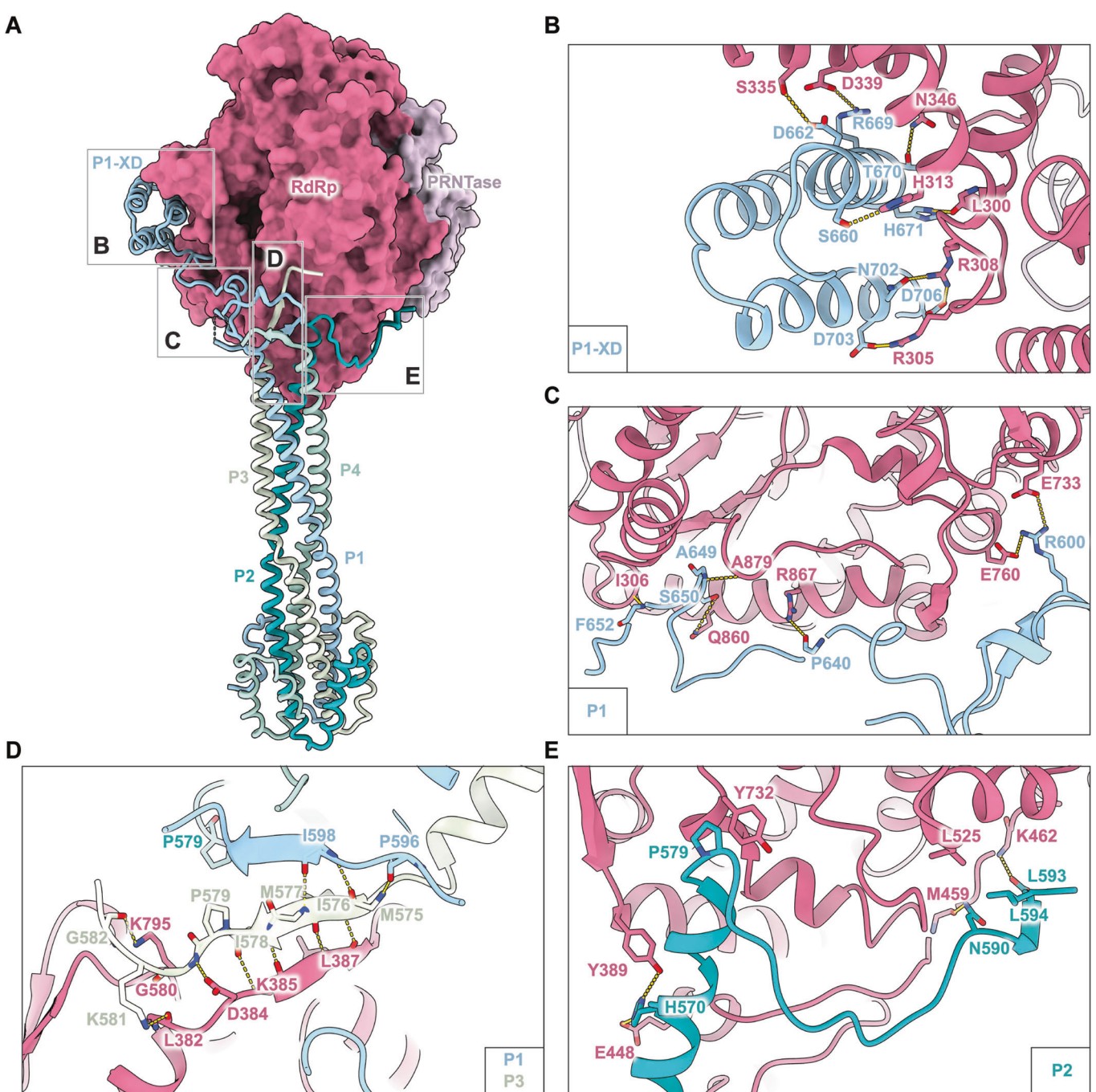

**Figure 4. Interactions of P-protein with the RdRp-domain of L-protein.**

(A) An overall representation of the L-P interaction sites, which can be divided into four different regions: B, C, D and E. (B) P1 forms a 3-helix bundle near the NTP entry site and interacts with the RdRp-domain through hydrophobic interactions, salt bridges, and hydrogen bonds. (C) The linker region of P2, located between the OD- and XD-domains, interacts with multiple residues on the RdRp-domain. (D) P3 forms a β-sheet, stabilized by interactions with P1 and L-protein at the L binding interface through hydrogen bonds and hydrophobic interactions. (E) Interactions between P2 and L-protein are mediated by salt bridges, hydrophobic interactions, and hydrogen bonds. The residues involved in hydrogen bond formation and hydrophobic interactions are shown as sticks. Hydrogen bonds and salt bridges are shown as yellow dashed lines.

the L-protein (Fig. 6A). The model was further extended by including a 21-nucleotide long 3′ leader sequence as the RNA template and product RNA of varying lengths, ranging from 9 to 15 nucleotides, along with the addition of NTPs and Mg ions. This new composite model predicts the pathway of the template RNA as it enters and subsequently exits the active site of the RdRp-domain. It also predicts how the product RNA egresses towards the PRNTase- and CD-domains (Fig. 6A). The model illustrates how the template RNA is positioned in the active site, where nucleotides are incorporated into the elongating nascent RNA, resulting in the

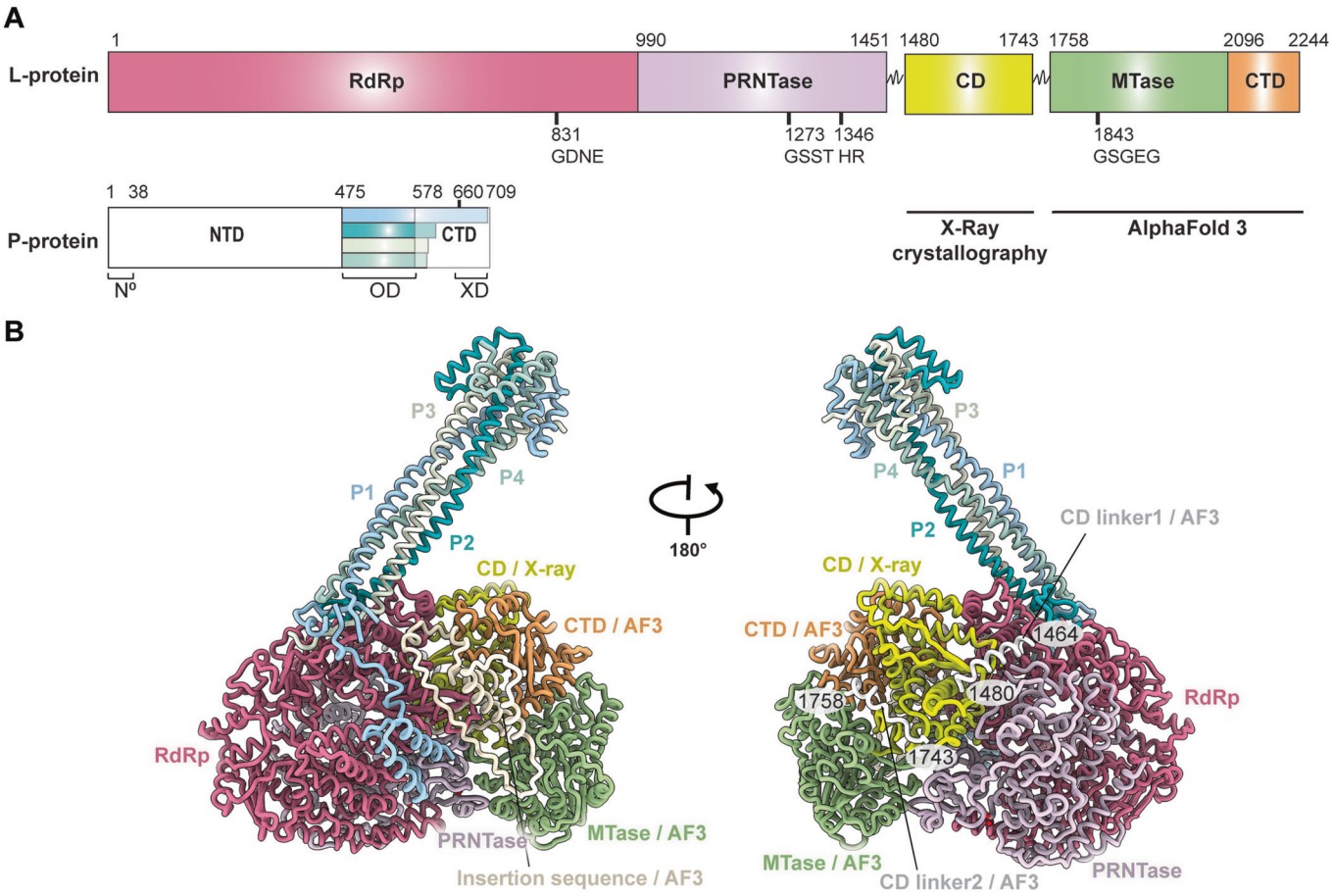

**Figure 5. A model for the complete NiV L-P complex.**

(**A**) Schematic diagram of L- and P-proteins with the indication that the structure of CD-domain was determined by X-ray crystallography while AF3 was used to predict the structures of the MTase- and CTD-domains. (**B**) Composite model of the NiV L-P complex. The RdRp- and PRNTase-domains are aligned with the full-length AF3 model of the L-protein, achieving a root-mean-square deviation (RMSD) of 1.7 Å. In addition, the CD-domain of the L-protein is aligned with the AF3 model, resulting in an RMSD of 0.9425 Å. The insertion sequence, which is disordered in the cryo-EM map, is predicted by AF3 and is highlighted in light cream colour. In addition, the residues for the CD linker1 (residues 1464–1480), which connects the PRNTase-domain to the CD-domain, and the CD linker2 (residues 1743–1758), which connects the CD-domain to the MTase-domain, are shown. The missing regions for these linkers are predicted by AF3 and represented in white colour.

formation of an intermediate RNA duplex (Fig. 6B). Following this, the two strands are separated, and the template and product are directed towards the template and product exit channels, respectively (Fig. 6B). The model also demonstrates the positioning of the conserved GxxT motif ([1273]GSST[1276]) of the priming loop, which is thought to bind GTP (Li et al, 2008; Liang et al, 2015; Te Velthuis et al, 2021) and the localization of the HR motif ([1347]HR[1348]), both of which are crucial for capping of the 5′ end of the nascent mRNA (Fig. 6C) (Ogino and Green, 2019). Our composite model also predicts the position of the crystallographically determined Mg1 ion bound to the α2 helix of the CD-domain, suggesting a role in positioning the GTP required for the capping process (Fig. 6D). To test whether this Mg ion plays a catalytic role in capping, we mutated the Mg1 binding residues (Mg1-AAA: S1523A, N1526A, S1529A) in a minigenome assay and an in vitro polymerase assay. In addition, we tested double mutants, HR-AA of the HR motif and GT-AA of the GxxT motif. The Mg1-AAA mutant completely abolished minigenome activity (Fig. 6E) while leaving in vitro polymerase activity unaffected (Fig. 6F). Both the

HR-AA and GT-AA double mutants also eliminated minigenome activity (Fig. 6E), likely due to disrupted covalent bond formation in the HR motif and impaired GTP binding in the GxxT mutant. Notably, while HR-AA did not affect polymerase activity in vitro on the short leader template, GT-AA abolished RNA synthesis (Fig. 6F). Leader sequences are not capped, however synthesis initiation of RNA products depends on efficient nucleotide positioning by the priming loop within the active site, which could be impeded by mutations in the GxxT motif. These structural and functional insights underscore the intricate coordination of multiple domains within the polymerase complex necessary for RNA processing.

## Discussion

This study provides structural and functional insights into the NiV polymerase complex. Although the sequences of L-protein differ across the nsNSVs (Fig. EV3; Appendix Fig. S1), the architecture

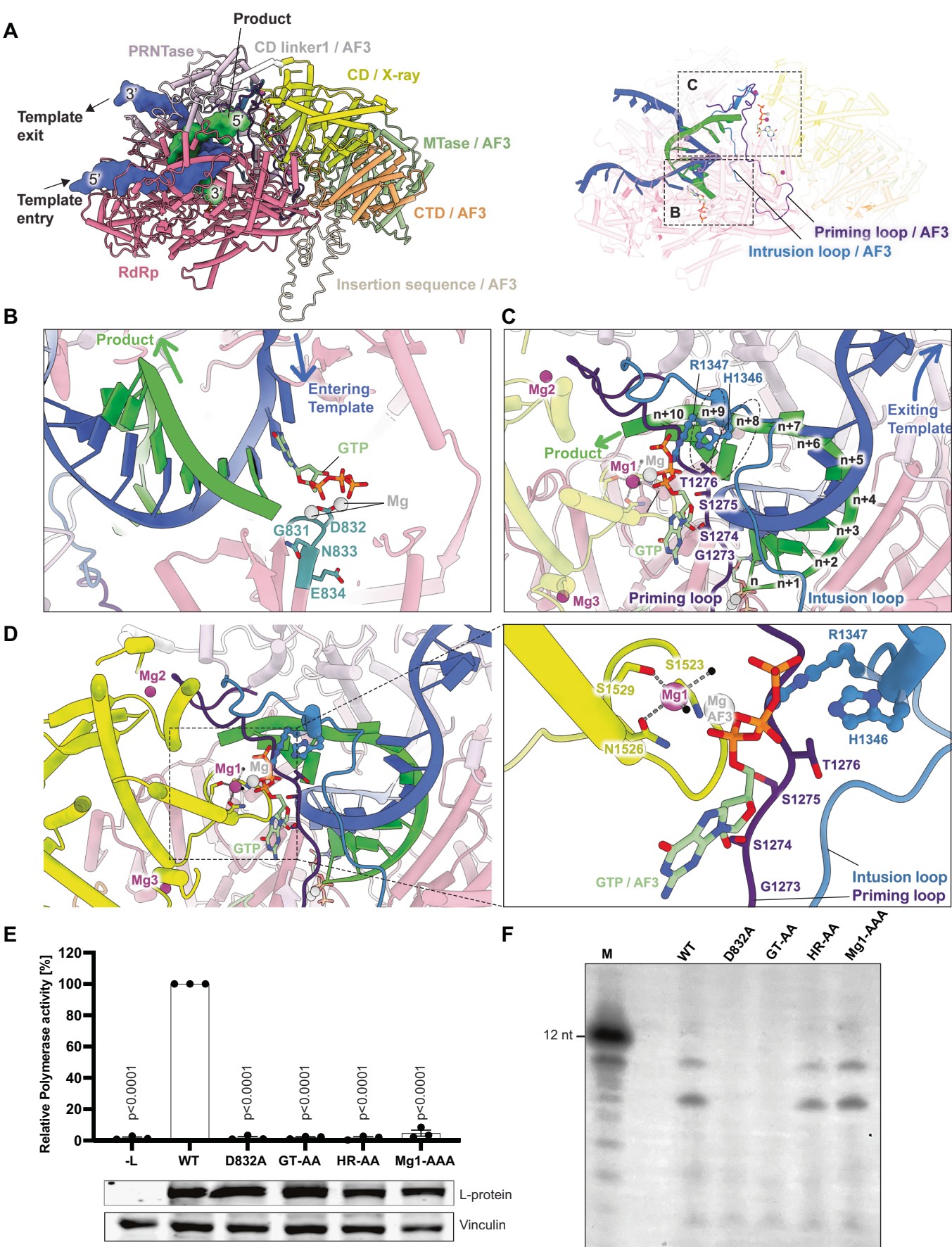

◀ **Figure 6. Modelling of RNA on the composite L-P model.**

(A) The RNA template with a growing nascent RNA is modelled on the composite L-P structure. The template and product RNA are represented by surface in 5′-3′ direction. A cartoon representation of the template and the product RNAs is shown on the right. Template RNA used in the AF3 composite models is 5′-UUUGUUGUUAACGCAAAAAAA-3′ and product RNA is 5′-AAAUUGCGUUA-3′. (B) A close-up view of the AF3 model of the active site pocket where the product RNA is synthesized, highlighting the incorporation of nucleotides complementary to the template RNA. This process results in the formation of an intermediate duplex RNA. The active site residues are shown as sticks, and the Mg ions are in grey. (C) A close-up view that shows the number of nucleotides that need to be added to the RNA product to reach the HR motif for the covalent attachment of the RNA to the L-protein. GTP, the $^{1347}HR^{1348}$ motif and the $^{1273}GSST^{1276}$ motif of the AF3 model are depicted as sticks, while Mg1 ion bound to the CD-domain (yellow) in the crystal structure is represented in magenta. (D) The crystal structure of CD-domain bound to Mg1 is shown. Water molecules coordinating Mg1 are represented as black spheres. A focused view illustrates the approximate position of the Mg1 in relation to the predicted coordination with GTP, as predicted by AF3. (E) Minireplicon assay was performed to evaluate the effect of point mutations in the RdRp-domain active site (D832A), the GxxT motif (GT-AA: G1273A, T1276A), the HR motif (HR-AA: H1347A, R1348A), or the Mg1 binding residues (Mg1-AAA: S1523A, N1526A, S1529A) of L-protein on polymerase activity. BSR-T7/5 cells were transfected with plasmids encoding L- (wild type or mutant), N- and P-proteins, together with a minigenome encoding for the Gaussia luciferase. Relative Gaussia luciferase activities are presented as the mean percentage activity of WT (±SEM), $n = 3$ independent biological replicates of technical duplicates. $P$ values from Dunnett's multiple comparison (WT) one-way ANOVA test are indicated. To assess the expression levels of L-proteins, HEK-293T cells were transfected with pcDNA3.1 plasmids expressing for L-protein (wild type or mutant). At 24 h p. t., expression levels were analysed by western blot using the indicated antibodies. (F) In vitro [α32P]-GTP incorporation assays for the 3′ extension activity of the L-P complex assessing elongation using a 12 nucleotide(nt)-long NiV leader sequence as a template, a 4-mer primer and purified NiV L-protein (wild type or mutant). A radiolabelled 12-mer RNA sequence served as a marker (M), lengths are indicated on the left. Source data are available online for this figure.

and key structural motifs of the L-P complex are conserved (Figs. EV3 and EV4). Structural comparison of the RdRp- and PRNTase-domains across paramyxoviral, pneumoviral, filoviral, and rhabdoviral L-proteins yield valuable insights into evolutionary adaptations and species-specific variations in polymerase functionality (Figs. EV3 and EV4). The RdRp-domain, which plays a central role in RNA synthesis, exhibits high structural conservation among nsNSVs (Figs. EV3 and EV4A). However, despite this conservation, the RdRp-domain can accommodate different insertions, such as the palm insertion observed in NiV, and the insertion in the N-terminal region of the RdRp-domain in EBOV (Yuan et al, 2022). These insertions likely contribute to the functional diversity and regulatory mechanisms observed across the different viral species. In addition, the PRNTase-domain, demonstrates significant flexibility in particular functional loops, such as the priming and intrusion loops (Fig. EV4B) in performing specific functions during replication and transcription. During the submission of this paper, additional structures of the NiV L-P complex were also made available online (Hu et al, 2024; Yang et al, 2024), revealing a conformation similar to our own NiV L-P complex structure, including the conformations of the RdRp- and PRNTase-domains and the binding of P-protein to the L-protein. However, the NiV L-P structure presented here provides distinct insights on structure and function within the broader context of the complete polymerase, as discussed in detail below.

RNA synthesis by the NiV polymerase complex is a tightly regulated process involving coordinated interactions between the L-protein, P-protein, and the RNA template encapsidated by N-protein. The 3′ end of the genome is composed of the leader RNA region which is a bipartite promoter that is highly conserved among paramyxoviruses. This sequence is recognized by the L-protein to initiate RNA synthesis (Jordan et al, 2018). After binding to the leader RNA, the polymerase complex initiates and either proceeds with the replication of full-length antigenomes and genomes or releases the short initial leader RNA product and scans through the genome until it locates one of the 'gene start' elements at the beginning of each gene to initiate transcription and produce mRNA. Both replication and transcription use an unprimed initiation mechanism, however, replication initiates terminally as opposed to transcription which initiates internally.

Based on our modelling, the active site can accommodate a 9-mer intermediate duplex formed during RNA synthesis, before the two strands—template and product—are separated and directed towards their respective exit channels (Fig. 6B).

Structural studies of L-P complexes from other viruses have revealed the conformation of the CD-domain along with the presence of the C-terminal domains (Abdella et al, 2020; Cong et al, 2023; Horwitz et al, 2020; Li et al, 2024; Xie et al, 2024). These studies have demonstrated that the positioning of the CD-domain modulates the organization of the MTase- and CTD-domains, thereby allowing the RNA to access the MTase-domain (Li et al, 2024; Xie et al, 2024). The nascent transcript RNA is initially capped at its first nucleotide, with the cap subsequently methylated by the MTase (Abraham et al, 1975). This process involves the covalent attachment of the first nucleotide of the product mRNA to the side chain of the conserved histidine residue in the HR motif, forming an intermediate (L)-(histidyl-Nε2)–pRNA (L-pRNA) (Ogino and Green, 2019). Following the formation of the L-pRNA complex, RNA is transferred from L-pRNA to GDP to form the cap structure. The MTase-domain subsequently methylates the N7 and O2 of the newly formed cap. Our model suggests that a 9 to 10 nts long product is required to be synthesized to reach the conserved HR motif ($^{1347}HR^{1348}$) for subsequent covalent attachment (Fig. 6C). In addition, the model predicts the binding of GTP to the conserved GxxT motif ($^{1273}GSST^{1276}$) in the priming loop (Fig. 6C,D). The transfer of the RNA onto the GDP is metal-dependent mainly involving divalent cations (Ogino et al, 2010). The crystal structure of the CD-domain reveals the presence of three bound Mg ions. Superimposition of the CD-domain with the AF3 model ideally positions the flexible α2 helix of the CD-domain, and the bound Mg1 close to the predicted GTP, suggesting a crucial role in the capping process by aiding the positioning of the GTP during the transfer reaction of PRNTase-domain (Fig. 6D). Gratifyingly, our functional studies reveal that mutating the residues that coordinate Mg1 lead to a complete loss of polymerase activity in a minigenome assay but do not affect in vitro RNA polymerisation showing that the RdRp-domain is still functional in synthesizing the RNA. Similarly, mutations in the HR and GxxT motifs abolish polymerase function in a minigenome assay but differ in the in vitro assays where, like the Mg mutants, the HR mutants retain in vitro activity. This suggests a direct catalytic role for the Mg1 ion in the synthesis of the cap structure during transcription and confirms the role of the HR motif in the covalent binding of the newly synthesised RNA. These

findings unveil a critical, yet previously unknown function of the CD-domain in coordinating a catalytic Mg ion essential for RNA synthesis and capping.

In addition, the crystal structure of the CD-domain has another bound Mg ion (Mg2) which engages with residues involved in crystal contacts (Appendix Fig. S8A,B) but—in the context of the full-length polymerase—could have a structural role stabilizing the capping conformation by stabilizing the CD linker1 (Appendix Fig. S8C-E). This comprehensive model advances our understanding of the spatial arrangement and functional dynamics within the NiV polymerase complex, providing a detailed framework for further investigations into viral replication mechanisms.

Based on observations of tetramers of the P-protein in nsNSV L-P complexes, both the cartwheeling and sliding models have been proposed to describe the movement of the polymerase complex along the nucleocapsid (Curran, 1998; Du Pont et al, 2019; Kolakofsky, 2016; Kolakofsky et al, 2004; Kolakofsky et al, 2021). The cartwheeling model implies that all four XD-domains must bind sequentially to L-protein during RNA synthesis. Recent studies have challenged this model by showing that tetramers of the P-protein where one to three XD-domains are deleted, maintain comparable or even heightened RNA synthesis activity (Du Pont et al, 2019). Notably, even a single XD-domain capable of binding to N-protein is adequate for minigenome transcription (Du Pont et al, 2019). In addition, our structural studies reveal that around 3500 Å² of surface area of the L-protein are bound by P-protein,

suggesting that the dissociation of P-protein from L-protein is highly unlikely. These observations lend support to the alternative sliding model where the oligomeric P-protein moves along the nucleocapsid without necessitating rotational movement. In the case of NiV, it is likely that the sliding of the L-P complex along the nucleocapsid is driven primarily by the process of RNA synthesis itself (Appendix Fig. S9). This movement is facilitated by the continuous association of the polymerase complex with the RNA strand, ensuring that nucleotides are added to the growing RNA chain. This continuous contact is essential for efficient transcription and replication, stabilizing the polymerase complex and enhancing processivity, ultimately ensuring the fidelity and efficiency of viral RNA synthesis.

NiV infections continue to pose a threat to public health. While a vaccine targeting the related HeV has proven effective in preventing zoonotic transmission through horses in Australia (Middleton et al, 2014), the primary reservoir for NiV—fruit bats—renders a similar strategy less feasible for controlling NiV outbreaks. This structural and functional characterization of the NiV L-P complex provides the basis for understanding the molecular details and the function of the polymerase complex and thus accelerates the development of therapeutic antiviral drugs active against the NiV polymerase complex.

# Methods

**Reagents and tools table**

| Reagent/Resource | Reference or Source | Identifier or Catalog Number |
|---|---|---|
| **Experimental models: cell lines** | | |
| HEK-293T (Human embryonic kidney 293T) cells | Sir William Dunn School of Pathology Oxford, Cellbank | #4454 |
| BSR-T7/5 cells (derived from baby hamster kidney cells (BHK-21)) | Sir William Dunn School of Pathology Oxford, Cellbank | #12053 RRID_CVCL_RW96 |
| Sf9 (Spodoptera frugiperda) cells | Thermo Fisher Scientific | Cat No: 11496015 |
| **Recombinant DNA** | | |
| pCAGGs-NiV-L | Bruhn et al, 2019 | |
| pCAGGs-NiV-P | Bruhn et al, 2019 | |
| pCAGGs-NiV-N | Bruhn et al, 2019 | |
| pUC57-BMG | Bruhn et al, 2019 | |
| pcDNA3.1-NiV-L | This study | |
| pcDNA3.1-NiV-P | This study | |
| pcDNA3.1-NiV-N | This study | |
| pcDNA3.1-Gluc1-NiV-L | This study | |
| pcDNA3.1-Gluc2-empty | Chen et al, 2019 | |
| pcDNA3.1-Gluc2-NiV-P | This study | |
| pcDNA3.1-Gluc2-NiV-P-I705A | This study | |
| pcDNA3.1-Gluc2-NiV-P-N702A | This study | |
| pcDNA3.1-Gluc2-NiV-P-H671A | This study | |
| pcDNA3.1-Gluc2-NiV-P-T670A | This study | |
| pcDNA3.1-Gluc2-NiV-P-R669A | This study | |

| Reagent/Resource | Reference or Source | Identifier or Catalog Number |
|---|---|---|
| pcDNA3.1-Gluc2-NiV-P-S660A | This study | |
| pcDNA3.1-Gluc2-NiV-P-A649G | This study | |
| pcDNA3.1-Gluc2-NiV-P-P640A | This study | |
| pcDNA3.1-Gluc2-NiV-P-R600A | This study | |
| pcDNA3.1-Gluc2-NiV-P-L594A | This study | |
| pcDNA3.1-Gluc2-NiV-P-E593A | This study | |
| pcDNA3.1-Gluc2-NiV-P-N591A | This study | |
| pcDNA3.1-Gluc2-NiV-P-N590A | This study | |
| pcDNA3.1-Gluc2-NiV-P-S573A | This study | |
| pcDNA3.1-Gluc2-NiV-P-H570A | This study | |
| pcDNA3.1-Gluc2-NiV-P-S565A | This study | |
| pcDNA3.1-Strep-Strep-Flag-NiV-P-I705A | This study | |
| pcDNA3.1- Strep-Strep-Flag -NiV-P-N702A | This study | |
| pcDNA3.1- Strep-Strep-Flag -NiV-P-H671A | This study | |
| pcDNA3.1- Strep-Strep-Flag -NiV-P-T670A | This study | |
| pcDNA3.1- Strep-Strep-Flag -NiV-P-R669A | This study | |
| pcDNA3.1- Strep-Strep-Flag -NiV-P-S660A | This study | |
| pcDNA3.1- Strep-Strep-Flag -NiV-P-A649G | This study | |
| pcDNA3.1- Strep-Strep-Flag -NiV-P-P640A | This study | |
| pcDNA3.1- Strep-Strep-Flag -NiV-P-R600A | This study | |
| pcDNA3.1- Strep-Strep-Flag -NiV-P-L594A | This study | |
| pcDNA3.1- Strep-Strep-Flag -NiV-P-E593A | This study | |
| pcDNA3.1- Strep-Strep-Flag -NiV-P-N591A | This study | |
| pcDNA3.1- Strep-Strep-Flag -NiV-P-N590A | This study | |
| pcDNA3.1- Strep-Strep-Flag -NiV-P-S573A | This study | |
| pcDNA3.1- Strep-Strep-Flag -NiV-P-H570A | This study | |
| pcDNA3.1- Strep-Strep-Flag -NiV-P-S565A | This study | |
| pCAGGs-NiV-L-Del602-657 | This study | |
| pCAGGs-NiV-L- Del668-710 | This study | |
| pCAGGs-NiV-L-Del602-657.Del668-710 | This study | |
| pcDNA3.1-NiV-L-D832A | This study | |
| pcDNA3.1-NiV-L- H1347A, R1348A | This study | |
| pcDNA3.1-NiV-L- G1273A, T1276A | This study | |

| Reagent/Resource | Reference or Source | Identifier or Catalog Number |
|---|---|---|
| pcDNA3.1-NiV-L- S1523A, N1526A, S1529A | This study | |
| pFL-NiV-L-P | This study | |
| pFL-NiV-L-D832A | This study | |
| pFL-NiV-L- G1273A, T1276A | This study | |
| pFL-NiV-L- H1347A, R1348A | This study | |
| pFL-NiV-L- S1523A, N1526A, S1529A | This study | |
| **Antibodies** | | |
| Rabbit anti-L | Thermo Fisher Scientific | |
| Mouse anti-Flag | Sigma Aldrich | RRID: AB_259529 |
| Mouse anti-myc | Sigma Aldrich | RRID: AB_439694 |
| Rabbit anti-GAPDH | Cell Signalling Technology | RRID: AB_561053 |
| Rabbit anti-Vinculin | Abcam | RRID: AB_11144129 |
| Goat anti-mouse IgG-800CW | LI-COR Bioscience | 926-32210 |
| Goat anti-rabbit IgG-800CW | LI-COR Bioscience | 926-32211 |
| **Oligonucleotides and other sequence-based reagents** | | |
| L_CD_ BamHI_1480_F_pET28a-Sumo | For bacterial expression of recombinant CD-domain protein | CGCGGATCC ATGCTGGACTTCCCCCTGT |
| L_CD_ HindIII_1742_R_pET28a-Sumo | For bacterial expression of recombinant CD-domain protein | CCCAAGCTT CTA TTGACGGATACGCAGCTG |
| L protein – fwd-NotI w/o ATG: | Sigma Aldrich | TAAGCAGCGGCCGCGCTGATGAACTGAGTATTTCCGAC |
| L protein – fwd-NotI: | Sigma Aldrich | TAAGCAGCGGCCGCATGGCTGATGAACTGAGTATTTCCGAC |
| L protein – rev-XhoI-NheI: | Sigma Aldrich | TAAGCACTCGAGGCTAGCTCAGATAATGGAGATGTAGCCGATAATCTTCC |
| L protein – rev-XhoI w/o Stop: | Sigma Aldrich | TAAGCACTCGAGGATAATGGAGATGTAGCCGATAATCTTCC |
| P protein fwd NotI w/o ATG: | Sigma Aldrich | TAAGCAGCGGCCGCGACAAACTGGAACTGGTCAACGAC |
| P protein fwd NotI: | Sigma Aldrich | TAAGCAGCGGCCGCATGGACAAACTGGAACTGGTCAACG |
| P protein rev XhoI w/o Stop: | Sigma Aldrich | TAAGCACTCGAGGATATTTCCATCAATGATGTCGTTCACG |
| N protein fwd NotI: | Sigma Aldrich | TAAGCAGCGGCCGCATGTCAGATATTTTTGAAGAAGCAGCATCC |
| N Protein rev Xho: | Sigma Aldrich | TAAGCACTCGAGTCACACATCGGCCCTGACGAAG |
| pcDNA-NiV-L-D832A fwd | Sigma Aldrich | CATTGCTGCAATCGTGCAGGGGGCAAACGAAAGC |
| pcDNA-NiV-L-D832A rev | Sigma Aldrich | GTGATGGCAATGCTTTCGTTTGCCCCCTGCAC |
| pcDNA-NiV-L-1347AA fwd | Sigma Aldrich | CAATCTGTCTGCCGCACTGCGGGACAAAAGTACCCAG |
| pcDNA-NiV-L-1347AA rev | Sigma Aldrich | GTCCCGCAGTGCGGCAGACAGATTGTTACTTGTAGACAC |
| pcDNA-NiV-L-GxxT-AA-fwd | Sigma Aldrich | CCATATGTGGCCTCCTCTGCTGACGAAAGATCTGATATCAAGC |
| pcDNA-NiV-L-GxxT-AA-rev | Sigma Aldrich | CTTTCGTCAGCAGAGGAGGCCACATATGGGACCCGGATGCTTG |
| Gib-pcDNA-NiV-L-Mg2-AAA-rev | Sigma Aldrich | GGCATTGATGGCATCGTCTGCGTCGATGGCCAGATGCTGTTTC |
| Gib-pcDNA-NiV-L-Mg2-AAA-fwd | Sigma Aldrich | CAGACGATGCCATCAATGCCCTGATTACCGAGTTTCTGATTG |
| Gibson-pcDNA-NiV-L-Del 1 fwd | Sigma Aldrich | TGAGCATCAGCTACCACAATATGTCTCCTAACATCCGGAACAG |
| Gibson-pcDNA-NiV-L-Del 1 rev | Sigma Aldrich | CATATTGTGGTAGCTGATGCTCAGCTGGAACAGGGTCTTCAG |
| Gibson-pcDNA-NiV-L-Del 2 fwd | Sigma Aldrich | ATCCGGAACAAGTTTGATACCGTGAGCGCCTTCCTG |
| Gibson-pcDNA-NiV-L-Del 2 rev | Sigma Aldrich | GGTATCAAACTTGTTCCGGATGTTAGGAGACATATTGTGGTAC |
| Gibson-pcDNA-NiV-L-Del 1-2 fwd | Sigma Aldrich | CAATATGTCTCCTAACATCCGGAACAAGTTTGATACCGTGAGCGCCTTCCTGACAACTG |

| Reagent/Resource | Reference or Source | Identifier or Catalog Number |
|---|---|---|
| Gibson-pcDNA-NiV-L-Del 1-2 rev | Sigma Aldrich | CGGATGTTAGGAGACATATTGTGGTAGCTGATGCTCAGCTGGAACAGGGTCTTCAG |
| Gibson-pcDNA-NiV-L-Sac1 fwd | Sigma Aldrich | TTTGGCAAAGAATTCGAGCTCATGGCTGATGAACTGAGTATT |
| Gibson-pcDNA-NiV-L-Xho rev | Sigma Aldrich | GCAGAGGGAAAAAGATCCTCGAGTCAGATAATGGAGATGTAGCCG |
| pFL-NiV-L-D832A fwd | Sigma Aldrich | gctgctatcgtgcagggcGCAaacgagtc |
| pFL-NiV-L-D832A rev | Sigma Aldrich | ctgggtgatagcgatagactcgttTGCgccctg |
| pFL-NiV-L-H1347AA fwd | Sigma Aldrich | cctctaacaacctgtccGCcGCtctgcgcgacaag |
| pFL-NiV-L-H1347AA rev | Sigma Aldrich | gaactgggtagacttgtcgcgcagaGCgGCggacaggttg |
| pFL-NiV-GxxT-AA-fwd new | Sigma Aldrich | ctacgtgGCCtcctccGCCgacgagcgttctg |
| pFL-NiV-GxxT-AA-rev new | Sigma Aldrich | GGCggaggaGGCcacgtagggcacacggatgg |
| Gib-pFL-NiV-L-Mg2-AAA-rev | Sigma Aldrich | GGCgttgatGGCgtcgtcAGCgtcgatagccaggtgctgcttc |
| Gib-pFL-NiV-L-Mg2-AAA-fwd | Sigma Aldrich | ctgacgacGCCatcaacGCCctcatcaccgagttcctcatcg |
| NiV-P-N702A fwd | Sigma Aldrich | CAAATACCGTGGCCGACATCATTGATGGAAATATC |
| NiV-P-N702A rev | Sigma Aldrich | CAATGATGTCGGCCACGGTATTTGCGATC |
| NiV-P-H671A fwd | Sigma Aldrich | GTCAAAACACTGATTAGAACTGCAATCAAGGAC |
| NiV-P-H671A rev | Sigma Aldrich | CAGTTCTCGGTCCTTGATTGCAGTTCTAATC |
| NiV-P-T670A fwd | Sigma Aldrich | CACTGATTAGAGCTCACATCAAGGACCGAG |
| NiV-P-T670A rev | Sigma Aldrich | CCTTGATGTGAGCTCTAATCAGTGTTTTGACCAC |
| NiV-P-R669A fwd | Sigma Aldrich | GATGTGGTCAAAACACTGATTGCAACTCACATC |
| NiV-P-R669A rev | Sigma Aldrich | CTCGGTCCTTGATGTGAGTTGCAATCAGTG |
| NiV-P-S660A fwd | Sigma Aldrich | CCAATGGCTGACGATAGTGCACGGGATG |
| NiV-P-S660A rev | Sigma Aldrich | GTTTTGACCACATCCCGTGCACTATCGTC |
| NiV-P-R600A fwd | Sigma Aldrich | GATTGGGGCAGATGTCCTGGAGCAGCAGAG |
| NiV-P-R600A rev | Sigma Aldrich | CAGGACATCTGCCCCAATCACAGGTTTCAGCTCGG |
| NiV-P-N590A fwd | Sigma Aldrich | CAAGGGCAAAGCCAACCCCGAGCTGAAAC |
| NiV-P-N590A rev | Sigma Aldrich | CGGGGTTGGCTTTGCCCTTGCGTTCTCCTTTCC |
| NiV-P-P579A fwd | Sigma Aldrich | CATGATGATCATGATTGCAGGCAAGGGGAAAG |
| NiV-P-P579A rev | Sigma Aldrich | CCTTGCCTGCAATCATGATCATCATGGACAC |
| NiV-P-I578A fwd | Sigma Aldrich | CATGATGATCATGGCTCCAGGCAAGGGGAAAG |
| NiV-P-I578A rev | Sigma Aldrich | CTTGCCTGGAGCCATGATCATCATGGACACC |
| NiV-P-S573A fwd | Sigma Aldrich | GGGCCATCTGGTGGCCATGATGATCATGATTCCAG |
| NiV-P-S573A rev | Sigma Aldrich | GATCATCATGGCCACCAGATGGCCCTCGATTGTG |
| NiV-P-H570A fwd | Sigma Aldrich | CAATCGAGGGCGCTCTGGTGTCCATGATGATC |
| NiV-P-H570A rev | Sigma Aldrich | GGACACCAGAGCGCCCTCGATTGTGCTCAG |
| NiV-P-S565A fwd | Sigma Aldrich | CAATACAGCACTGGCCACAATCGAGGGCCATCTG |
| NiV-P-S565A rev | Sigma Aldrich | CCTCGATTGTGGCCAGTGCTGTATTGGTCTTAGCC |
| NiV-P-N591A fwd | Sigma Aldrich | GGCAAAAGCGCCCCCGAGCTGAAACCTGTG |
| NiV-P-N591 rev | Sigma Aldrich | CTCGGGGGGCGCTTTTGCCCTTGCGTTCTCCTTTC |
| NiV-P-E593A fwd | Sigma Aldrich | GCAACCCCGCACTGAAACCTGTGATTGGGC |
| NiV-P-E593A rev | Sigma Aldrich | GTTTCAGTGCGGGGTTGCTTTTGCCCTTGC |
| NiV-P-L594A fwd | Sigma Aldrich | CCCCGAGGCGAAACCTGTGATTGGGCGAG |
| NiV-P-L594A rev | Sigma Aldrich | CAGGTTTCGCCTCGGGGTTGCTTTTGCCCTTG |
| NiV-P-A649G fwd | Sigma Aldrich | GAAACTAACGGAAGCCAGTTTGTGCCAATGGC |
| NiV-P-A649G rev | Sigma Aldrich | CAAACTGGCTTCCGTTAGTTTCCTCGAAATTCAGC |
| NiV-P-P640A fwd | Sigma Aldrich | CTGATCCTGGCTGAGCTGAATTTCGAGGAAAC |

| Reagent/Resource | Reference or Source | Identifier or Catalog Number |
|---|---|---|
| NiV-P-P640A rev | Sigma Aldrich | ATTCAGCTCAGCCAGGATCAGGTCGCCTCGCAG |
| NiV-P-I705A fwd | Sigma Aldrich | CGTGAACGACATCGCTGATGGAAATATC |
| NiV-P-I705A rev | Sigma Aldrich | TTTCCATCAGCGATGTCGTTCACGGTATTTGC |
| **Chemicals, enzymes and critical commercial kits** | | |
| DMEM | Sigma Aldrich | # Cat: D6429 |
| FBS | Sigma Aldrich | # Cat: F9665 |
| G418/Geneticin | Roche | # Cat: 108321-42-2 |
| Sf-900™ II SFM medium | Thermo Scientific | # Cat: 10902096 |
| Not-I | NEB | # Cat: R3189S |
| Xho-I | NEB | # Cat: R0146S |
| Protease inhibitor cocktail | Roche | # Cat: 04693116001 |
| Superdex 75 16/600 | GE Healthcare | # Cat: 28989333, cytiva |
| Rnasin | Promega | # Cat: N211A |
| Remdesivir triphosphate | MedChemExpress | # Cat: HY-126303 |
| $[\alpha^{32}P]$ GTP | Revvity | # Cat: BLU006H250UC |
| Maximum Sensitivity Radioisotope film | Carestream® Kodak® BioMax® | # Cat: Z363006-50EA |
| LT-1 transfection reagent | MirusBio | # Cat: MIR 2305 |
| Lipofectamine™ 2000 transfection reagent | Thermo Fisher Scientific | # Cat: 11668019 |
| Renilla Luciferase Assay System | Promega | # Cat: E2820 |
| Gibson assembly kit | NEB | # Cat: E5510S |
| **Software** | | |
| Prism 10.2 | GraphPad Software | RRID:SCR_002798 |
| ImageJ 1.54f | https://imagej.net/ | RRID:SCR_003070 |
| CryoSparc v4.4.1 | Structura Biotechnology Inc. | RRID:SCR_016501 |
| ModelAngelo | https://github.com/3dem/model-angelo | |
| COOT | MRC Laboratory of Molecular Biology | RRID:SCR_014222 |
| MolProbity | Duke University | RRID:SCR_014226 |
| Chimera ChimeraX | UCSF | RRID:SCR_004097 RRID:SCR_015872 |
| AlphaFold 2 and AlphaFold 3 | DeepMind Technologies Ltd. | RRID:SCR_025454 RRID:SCR_025885 |
| **Instrumentation** | | |
| GLOMAX 20/20 luminometer | Promega | RRID: SCR_018613 |
| FLUOStar Omega plate reader | BMG Labtech | # Cat: 415-101-AFL |
| Odyssey® DLx Imaging System | LI-COR Bioscience | # Cat: 9142-00 |
| Vitrobot Mark VI | Thermo Fisher Scientific | VI |
| Krios microscope | Thermo Fisher Scientific | Titan Krios |

## Cells and plasmids

Human embryonic kidney 293T cells (HEK-293T) were maintained in Dulbecco's modified Eagle medium (DMEM) with 10% fetal bovine serum (FBS; Sigma Aldrich) at 37 °C and 5% $CO_2$. BSR-T7/5 cells, cells stably expressing the T7 polymerase that are derived from baby hamster kidney cells (BHK-21) (Buchholz et al, 1999) were maintained in DMEM with 10% FBS and 1 mg/ml G418/Geneticin (Roche) every third cell culture passage. The *Spodoptera frugiperda* Sf9 cells were maintained in Sf-900™ II SFM medium (Thermo Scientific) at 27 °C. All cells tested negative for mycoplasma contamination. Cell lines were not authenticated.

pCAGGs plasmids expressing NiV L-, P- and N-proteins (Bangladesh, 2004) as well as the pUC57 plasmid expressing T7-dependent NiV bicistronic minigenome (BMG) (Bruhn et al, 2019) were kindly provided by Michael Lo (CDC). Coding sequences for NiV L and N were amplified by PCR and subcloned into pcDNA3.1 expression vectors. For the NiV P-protein, the coding sequence was transferred into a pcDNA3.1 vector expressing an N-terminal Flag-Strep-Strep tag. The inserts were amplified by PCR and subcloned using NotI and XhoI restriction sites. For protein interaction studies, the NiV L- and P-proteins sequences were cloned into pcDNA3.1 vectors designed to express the two components of split Gaussia luciferase (GLuc). The N-terminal half of the GLuc (Gluc1) was fused to the N-terminus of NiV L, while the C-terminal half of the GLuc (Gluc2) was fused to the N-terminus of NiV P. L and P coding sequences were amplified by PCR and cloned into the pcDNA3.1 vector using NotI and XhoI restriction sites.

The coding sequences for the Malaysian strain NiV L-protein (NCBI: NP_112028.1) with an N-terminal twin-Strep tag and P-protein (NP_112022.1) with an N-terminal octa-His tag were codon-optimized for expression in insect cells. These sequences were synthesized by SynBio Technologies and cloned into a pFL plasmid, downstream of the polyhedrin and p10 promoters, respectively.

pCAGGs plasmids expressing truncated versions of L-proteins as well as the pFL and pcDNA3.1 plasmids expressing L with the triple mutant of the Mg1 binding residues (Mg1-AAA: S1523A, N1526A, S1529A), the double mutant of the HR motif (HR-AA: H1347A, R1348A), and the double mutant of the GxxT motif (GT-AA: G1273A, T1276A) were generated by Gibson assembly (NEB). Single mutations in pcDNA3.1-TAP-P and pcDNA3.1-Gluc2-P plasmids were introduced by site-directed mutagenesis.

All plasmid constructs were confirmed by Sanger sequencing.

## Protein expression and purification of NiV L-P complex

The preparation of Baculovirus stocks and protein expression were performed following the Bac-to-Bac manual (Invitrogen). Two litres of Sf9 cells expressing the L-P complex were harvested 72 h post infection. The cells were resuspended in Buffer A (50 mM HEPES pH 7.4, 500 mM NaCl, 10% vol/vol glycerol, 2 mM tris(2-carboxyethyl) phosphine (TCEP), 2 mM $MgCl_2$) which was further supplemented with 0.05% wt/vol n-octyl beta-d-thioglucopyranoside, one protease inhibitor cocktail tablet (Roche, cOmplete Mini, EDTA-free), 2 mM phenylmethylsulfonyl fluoride (PMSF), Benzonase, and RNase. Cells were lysed using a Dounce homogeniser and clarified with centrifugation. The lysate was incubated with the pre-equilibrated Strep-Tactin® XT Sepharose resin (IBA Lifesciences) for 3 h and resin was washed with the Buffer A. The NiV L-P complex was eluted with 50 mM Biotin in Buffer A and further purified with Superose 6 increase 10/300 size exclusion column (GE Healthcare) in Buffer A. The fractions eluting after the void volume (between 10 and 12 ml) were collected and concentrated to 0.5 mg/ml. The L-D832A mutant, the triple mutant of the Mg1-binding residues (Mg1-AAA: S1523A, N1526A, S1529A), the double mutant of the HR motif (HR-AA: H1347A, R1348A), and the double mutant of the GxxT motif (GT-AA: G1273A, T1276A), all in complex with P, were expressed in 0.5-liter cultures and purified using the same methods as the wild-type L-P complex.

## Protein expression and purification of CD-domain of NiV L-protein

Residues 1480–1742, encoding the CD-domain of L-protein, were cloned into a pET28a vector with an N-terminal hexa-His and Sumo tag. The construct was transformed into BL21(DE3) cells and protein expression was induced with 0.5 mM IPTG at $OD_{600} = 0.6$ and expression was carried out at 18 °C for 18 h. The bacterial cell pellet was lysed by sonication in a Lysis Buffer containing 50 mM HEPES pH 7.6, 500 mM NaCl, 10% vol/vol glycerol supplemented with protease inhibitors (Roche, cOmplete Mini, EDTA-free), RNase A, and lysozyme (Sigma). Supernatant cleared by ultra-centrifugation was filtered and loaded onto HisTrap 5 ml HP columns. Protein was washed and eluted with a gradient increase in imidazole concentration. Elution fractions were combined, desalted into the Final Buffer containing 20 mM HEPES pH 7.6, 500 mM NaCl, 10% vol/vol glycerol and then incubated with Ulp1 overnight at 4 °C. Cleaved protein was re-injected into HisTrap 5 ml HP to remove Ulp1 enzyme as well as the cleaved His-Sumo tag. The flowthrough was collected, concentrated, and injected into Super-dex 75 16/600 (GE Healthcare) that was pre-equilibrated with the Final Buffer. The protein was concentrated to 10 mg/ml and stored at −80 °C.

## In vitro RNA synthesis assay

For the NiV L-P complex in vitro assay, 3 µl reactions were set up containing reaction buffer [20 mM Tris pH 7.5, 10 mM KCl, 2 mM DTT, 0.5% Triton, 10% DMSO, 1 U Rnasin (Promega), 5 mM $MgCl_2$], 0.25 µM RNA-template derived from the NiV leader sequence (UGGUUUGUUCCC or UGGUCUGUUCCC), 0.5 µM recombinant L-P complex, 0.5 mM ATP, 0.5 mM CTP, 0.1 µM GTP and 200 µM primer (pACCA). ATP was substituted with remdesivir triphosphate (APExBIO), where indicated. The radioisotope tracer in these reactions was [α$^{32}$P] GTP (Revvity). The reactions were incubated at 30 °C for 1 h, stopped with the addition of 3 µl formamide loading buffer and denatured at 95$^{°}$C for 3 min. A $^{32}$P-5′ end labelled 12 nucleotide-long RNA served as a marker. RNA products were separated on a 22% polyacrylamide urea gel for 2.5 h at 35 W and the level of [α$^{32}$P] GMP incorporation was imaged by exposing a Maximum Sensitivity Radioisotope film (Carestream® Kodak® BioMax®) for 48 h.

## Minireplicon assay

BSR-T7/5 cells were seeded at a density of $5 \times 10^4$ cells per well in 24-well plates. 24 h later, cells were transfected in duplicates with the BMG plasmid, as well as pCAGGs or pcDNA3.1 plasmids expressing for NiV L-, (Strep-Strep-Flag-tagged) P-, N-proteins using LT-1 transfection reagent (MirusBio). For negative controls, NiV L-protein expressing plasmids were substituted with an empty vector, pCAGGs or pcDNA3.1, respectively. After 48 h, Gaussia luciferase activity was determined using the Renilla Luciferase Assay System (Promega). For this purpose, cells were lysed in 50 µl Renilla lysis buffer on a microplate shaker at room temperature. After 45 min, 20 µl of the cell lysates were mixed with 50 µl of the Renilla luciferase reagent and RLUs were analysed using a GLOMAX 20/20 luminometer (Promega).

## Split-luciferase assay

HEK-293T cells were seeded at a density of $4 \times 10^4$ cells per well in 96-well plates. 24 h later, cells were transfected in triplicates with 30 µg of pcDNA3.1-Gluc1-L-protein and pcDNA3.1-Gluc2-P-protein (wild type and mutant) using Lipofectamine™ 2000 transfection reagent (Thermo Fisher Scientific). For negative controls, pcDNA3.1-Gluc2-P-protein was substituted with pcDNA3.1-Gluc2-empty vector. After 24 h Gaussia luciferase signal was determined using the Renilla Luciferase Assay System (Promega). Cells were lysed in 30 µl Renilla lysis buffer on a microplate shaker at room temperature for 45 min. Then, 30 µl of the cell lysates were mixed with 30 µl of the Renilla luciferase reagent and RLUs were analysed using FLUOStar Omega plate reader (BMG Labtech).

## Western blot

HEK-293T cells were seeded at a density of $3 \times 10^5$ cells per well in 12-well plates. After 24 h, cells were transfected with pcDNA3.1, pcDNA3.1-Gluc (1/2) and pCAGGs plasmids expressing for NiV L- and P-proteins (wild type or mutant) using Lipofectamine™ 2000 transfection reagent (Thermo Fisher Scientific) according to the manufacturer's instructions. 24 h after transfection, cells were lysed with Radio-Immunoprecipitation Assay (RIPA) buffer [150 mM NaCl, 0.1% (w/v) SDS, 0.5% (w/v) sodium deoxycholate, 25 mM TRIS (pH 7.5), 1% (v/v) Triton X-100], supplemented with protease inhibitor cocktail (Roche, cOmplete, EDTA free). Clarified cell lysates were mixed with Laemmli buffer [25 mM TRIS-HCl (pH 6.8), 0.01% Bromophenol blue, 40% (v/v) Glycerol, 8% (w/v) SDS, 40 mM DTT], heated at 95 °C for 5 min, and proteins were resolved by SDS-PAGE and transferred to nitrocellulose membranes. L-proteins were detected using custom primary antibodies targeting the CD-domain of L-protein. These antibodies were generated by immunizing rabbits with purified CD-domain protein, with serum collected 72 days post-immunization (Thermo Fisher Scientific, 1:1000). P-proteins tagged with a Strep-Strep-Flag-tandem affinity purification (TAP) tag were detected using a mouse anti-Flag antibody (RRID: AB_259529, 1:1000), and P-proteins fused to Gluc2 were detected using a mouse anti-myc antibody (RRID: AB_439694, 1:1000). For loading controls, housekeeping proteins were detected using rabbit anti-GAPDH (RRID: AB_561053, 1:5000) and rabbit anti-Vinculin (RRID: AB_11144129, 1:2000). Primary antibodies were visualized with secondary antibodies: goat anti-rabbit IgG-800CW, or goat anti-mouse IgG-800CW (both from LI-COR Bioscience, 1:10,000). Protein bands were imaged using the Odyssey® DLx Imaging System (LI-COR Bioscience).

## Cryo-EM sample preparation, data acquisition and processing

3.5 µl of the NiV L-P complex at 0.5 mg/ml was applied to a freshly glow-discharged UltrAufoil Au 1.2/1.3 300 mesh grid. The sample was blotted for 6 s and plunge frozen in liquid ethane. All grids were prepared using a Vitrobot Mark VI (FEI) under conditions of 100% humidity and 20 °C. Cryo-EM data was collected at the Oxford Particle Imaging Centre (OPIC) using a 300 kV G3i Titan Krios microscope (Thermo Fisher Scientific) equipped with a SelectrisX energy filter and Falcon 4i direct electron detector. Automated data collection was setup in EPU 3.4 and a total of

22,859 movies were recorded in EER format, of which 14,319 were collected with a tilt angle of 30°. Data was collected using AFIS with a total dose of ~50 e-/Å², a calibrated pixel size of 0.7303 Å/pix, defocus range of −1.4 to −2.4, and with 10 eV slit. Cryo-EM data collection parameters and refinement statistics are summarized in Appendix Table S1.

Data processing was performed in CryoSparc v4.4.1 (Punjani et al, 2017) by following the workflow outlined in Fig. EV2. Briefly, motion correction and patch CTF estimations were performed for movie frames, and low-quality images were eliminated by manual inspection and excluded from further analysis. A template search was prepared using the PIV5 L-P structure (PDB 6V85) as the template. Following template picking, 2D classes were obtained, and the best-resolved classes were selected for ab initio model generation. The ab initio models were refined using Heterogenous refinement. The resulting optimal map was refined using NU-refinement with C1 symmetry. This refined map was then used to train Topaz picking. The picked particles were directly subjected to heterogeneous refinement, and the best resulting map was selected for further refinement using NU-refinement. This entire process, from particle picking to Topaz training, was repeated three times to increase the number of picked particles.

The map, containing 490,675 particles, was corrected for reference motion and subjected to 3D classification to identify classes that represented clear densities for the L-protein and XD-domain of the P-protein. A total of 299,261 particles were selected and refined for the L-protein using local NU refinements, resulting in a 2.52 Å resolution map. Meanwhile, 206,718 particles were refined for the tetramer of the P-protein using local NU refinement, yielding a 2.75 Å resolution map. The local maps were further processed with DeepEMhancer.

## Cryo-EM model building and refinement

Initial models were generated for L-protein and P tetramer separately using ModelAngelo (Jamali et al, 2024). To improve model geometry, multiple cycles of manual building were performed using COOT (Casanal et al, 2020) followed by real-space refinement against the corresponding local maps in PHENIX (Afonine et al, 2018). The model geometry was validated using MolProbity (Davis et al, 2007). Comprehensive statistics for the map and model are presented in the Appendix Table S1. Structural analysis and figure preparation were conducted using UCSF Chimera (Pettersen et al, 2004) and UCSF ChimeraX (Goddard et al, 2018).

## Crystallization of CD-domain, X-ray data collection and structure solving

Freshly purified CD-domain, at a concentration of 10 mg/mL, was mixed with a set of sitting-drop crystallization screens in ratios of 1:1, 2:1, and 1:2, and incubated at 20 °C. The crystals were obtained in 25 days from a condition consisting of 35% tert-Butanol and 0.1 M tri-Sodium Citrate pH 5.6. The crystals were harvested, cryo-preserved in the crystallization condition containing 20% v/v glycerol and flash frozen in liquid nitrogen.

X-ray diffraction data were collected on the I04 beamline at Diamond Light Source (Harwell, UK). The data processing was performed using the program DIALS in combination with XIA2 (Gildea et al, 2014). The structure was solved via PHASER

molecular replacement (McCoy et al, 2007), employing a search model generated by AlphaFold (Jumper et al, 2021). Subsequent manual model building was performed in COOT (Casanal et al, 2020), followed by refinement in PHENIX (Afonine et al, 2018). X-ray data collection and refinement statistics are summarized in Appendix Table S2.

## AlphaFold modelling

The RNA-bound NiV L-P complex structure was predicted in AlphaFold 3 by providing the leader and product RNA sequences as the input along with the corresponding protein sequences (Abramson et al, 2024). Specifically, the full-length sequence of NiV L-protein (UniProt ID: Q997F0) and four copies of the NiV P-protein (residues 479–709) (UniProt ID: Q9IK91) were used as inputs. In addition, the model included two Zn ions, four Mg ions, and three GTP nucleotides. The template RNA sequence used in the models was 5′-UUUGUUGUUAACGCAAAAAAA-3′, and the product RNA sequence was 5′-AAAUUGCGUUA-3′. For the X-ray crystallography analysis, a reference model of the CD-domain for molecular replacement was generated using AlphaFold 2 (Jumper et al, 2021), with residues 1480–1743 of the NiV L-protein as input.

## Quantification and statistics

Sample sizes were estimated on the basis of previous studies using similar methods and analyses (Bruhn et al, 2019; Jordan et al, 2018). For biochemical experiments, at least three independent replicates were performed, with all attempts at replication being successful. No data was excluded. The statistical analyses were performed with GraphPad Prism 10.2. software. Data represents mean values ± SEM. Statistical significance was assessed using appropriate methods. For experiments involving three or more groups, Dunnett's multiple comparison one-way ANOVA was conducted. Randomization or blinding was not relevant for our study.

## Data availability

Structural data generated in this study have been deposited in the Protein Data Bank (PDB) and the Electron Microscopy Data Bank (EMDB) under the following accession codes: PDB 9FTF (crystal structure of the CD, https://www.ebi.ac.uk/pdbe/entry/pdb/9ftf), PDB 9FUX (composite model of the NiV L-P complex, https://www.ebi.ac.uk/pdbe/entry/pdb/9fux). The corresponding EMDB entries are EMD-50781 (composite map, https://www.ebi.ac.uk/emdb/EMD-50781), EMD-50808 (consensus map, https://www.ebi.ac.uk/emdb/EMD-50808), EMD-50805 (L focused map, https://www.ebi.ac.uk/emdb/EMD-50805), and EMD-50807 (P focused map, https://www.ebi.ac.uk/emdb/EMD-50807). The AlphaFold 3 model of the NiV L-P-RNA complex was deposited in ZENODO with https://doi.org/10.5281/zenodo.13886435.

The source data of this paper are collected in the following database record: biostudies:S-SCDT-10_1038-S44318-024-00321-z.

## Peer review information

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

## Acknowledgements

We thank Michael Lo for providing the plasmids expressing NiV L, P, and N proteins and for the bicistronic minigenome (BMG) plasmid. We thank members of the Grimes and Fodor Laboratories for helpful comments and discussions. We thank Maria Harkiolaki for proofreading the main text. This work was supported by Pandemic Antiviral Discovery (PAD) Initiative INV-048922 (to J.M.G. and to E.F). Access to electron microscopes was provided by the OPIC Electron Microscopy Facility (funded by Wellcome JIF (060208/Z/00/Z) and equipment (093305/Z/10/Z) grants). Access to computational resources was supported by the Wellcome Trust Core Award Grant Number 203141/Z/16/Z with additional support from the NIHR Oxford BMRC. Molecular graphics and analyses were performed with UCSF Chimera, developed by the Resource for Biocomputing, Visualization, and Informatics at the University of California, San Francisco, with support from NIH P41-GM103311. The views expressed are those of the author(s) and not necessarily those of the NHS, the NIHR or the Department of Health.

## Author contributions

**Esra Balıkçı**: Conceptualization; Data curation; Formal analysis; Validation; Investigation; Visualization; Methodology; Writing—original draft; Writing—review and editing. **Franziska Güenl**: Data curation; Formal analysis; Validation; Investigation; Visualization; Methodology; Writing—original draft; Writing—review and editing. **Loïc Carrique**: Formal analysis; Supervision; Investigation; Methodology; Writing—review and editing. **Jeremy R Keown**: Formal analysis; Investigation. **Ervin Fodor**: Formal analysis; Supervision; Funding acquisition; Writing—original draft; Project administration; Writing—review and editing. **Jonathan M Grimes**: Conceptualization; Formal analysis; Supervision; Funding acquisition; Writing—original draft; Project administration; Writing—review and editing.

Source data underlying figure panels in this paper may have individual authorship assigned. Where available, figure panel/source data authorship is listed in the following database record: biostudies:S-SCDT-10_1038-S44318-024-00321-z.

## Disclosure and competing interests statement

The authors declare no competing interests.

# Expanded View Figures

**Figure EV1.  Purified NiV L-P complex is active.**

(**A**) Size exclusion chromatography profile of the L-P complex. Fractions eluting during the second peak (indicated by an arrow) were collected for subsequent functional and structural studies. An SDS-PAGE analysis of the purified protein complex is shown to the left of the chromatogram. (**B**) SDS-PAGE analysis of purified L-protein mutants, including the active site mutant (D832A), HR mutant (HR-AA: H1347A, R1348A), GT mutant (GT-AA: G1273A, T1276A), and Mg1 mutant (Mg1-AAA: S1523A, N1526A, S1529A), are shown. (**C, D**) In vitro [$\alpha^{32}$P]-GTP incorporation assays for the 3′ extension activity of the L-P complex assessing elongation with ($+$) or without ($-$) a 4-mer primer, a 12 nucleotide(nt)-long NiV leader sequence as a template, wild type NiV L-protein or the RdRp-domain active site mutant (L-D832A) (**C**). To evaluate the effect of remdesivir on RNA synthesis ATP was substituted with remdesivir triphosphate (**D**). A 12 nt sequence with a U-to-C mutation at position 5 (highlighted in orange) served as a template. A radiolabelled 12-mer RNA sequence served as a marker (M), lengths are indicated on the left. Source data are available online for this figure.

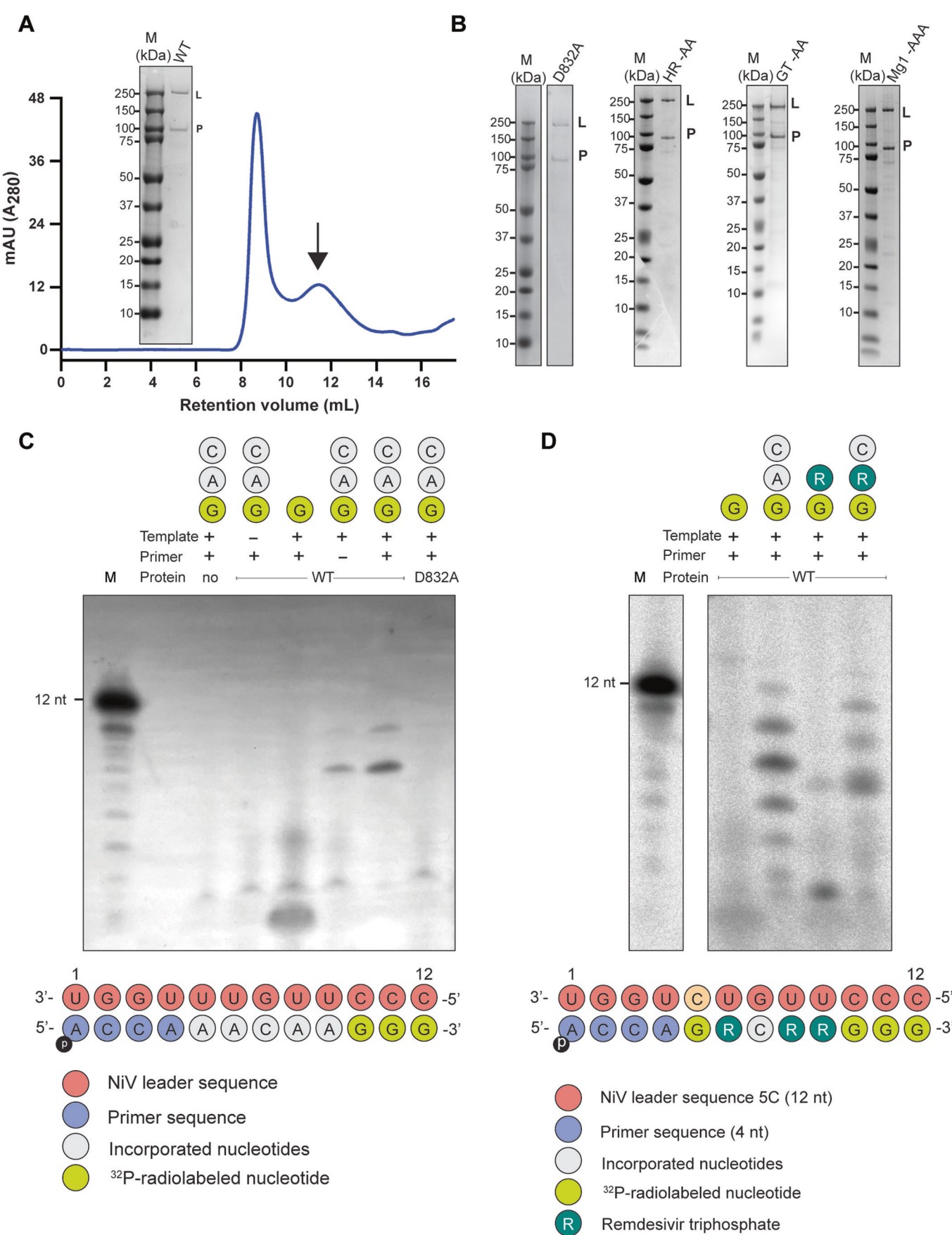

**NiV leader sequence**

**Primer sequence**

**Incorporated nucleotides**

**$^{32}$P-radiolabeled nucleotide**

**NiV leader sequence 5C (12 nt)**

**Primer sequence (4 nt)**

**Incorporated nucleotides**

**$^{32}$P-radiolabeled nucleotide**

**Remdesivir triphosphate**

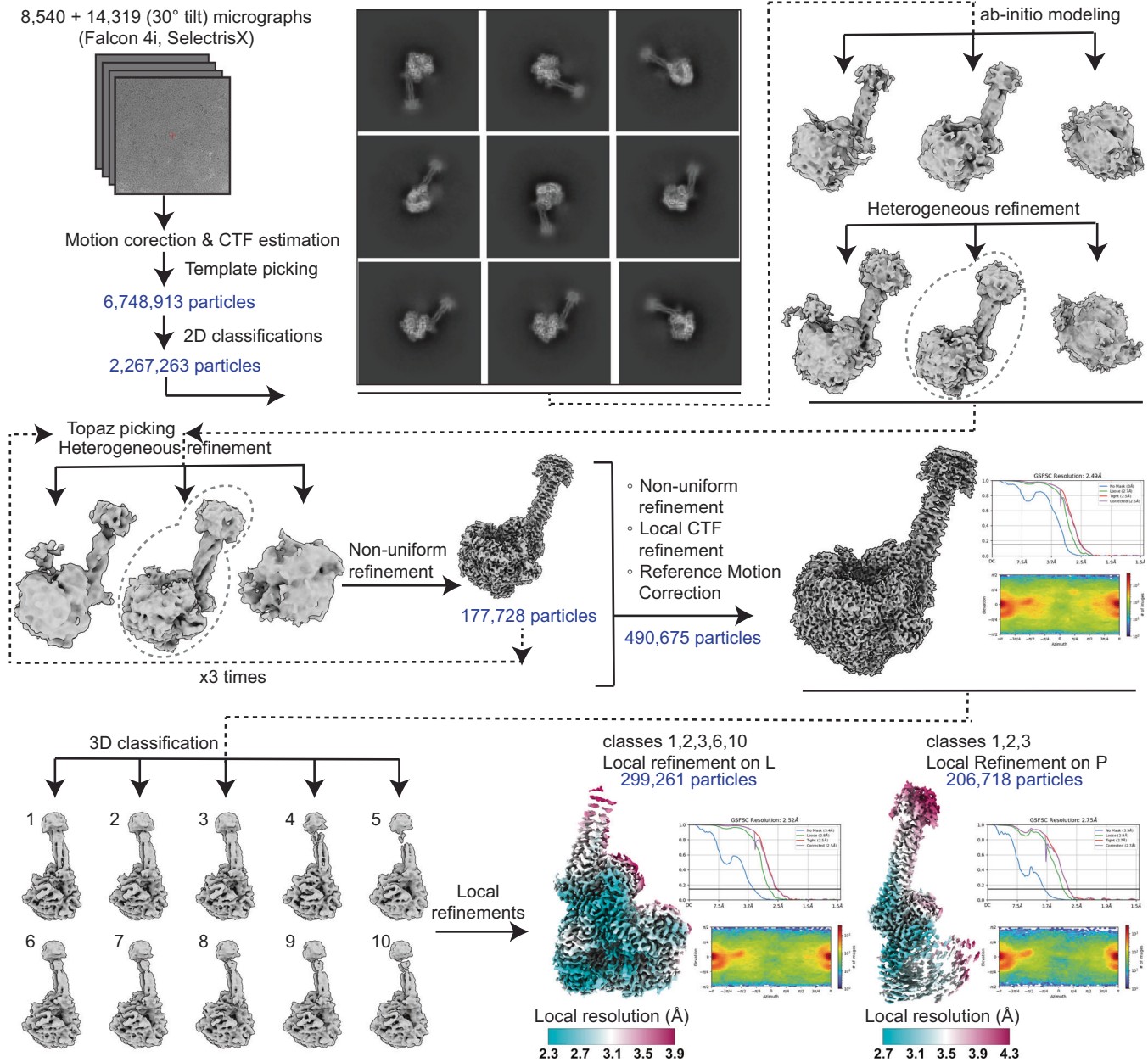

**Figure EV2. Cryo-EM data processing workflow.**

Flowchart indicates the path for data processing, classification and map reconstitution of the NiV L–P complex using CryoSPARC.

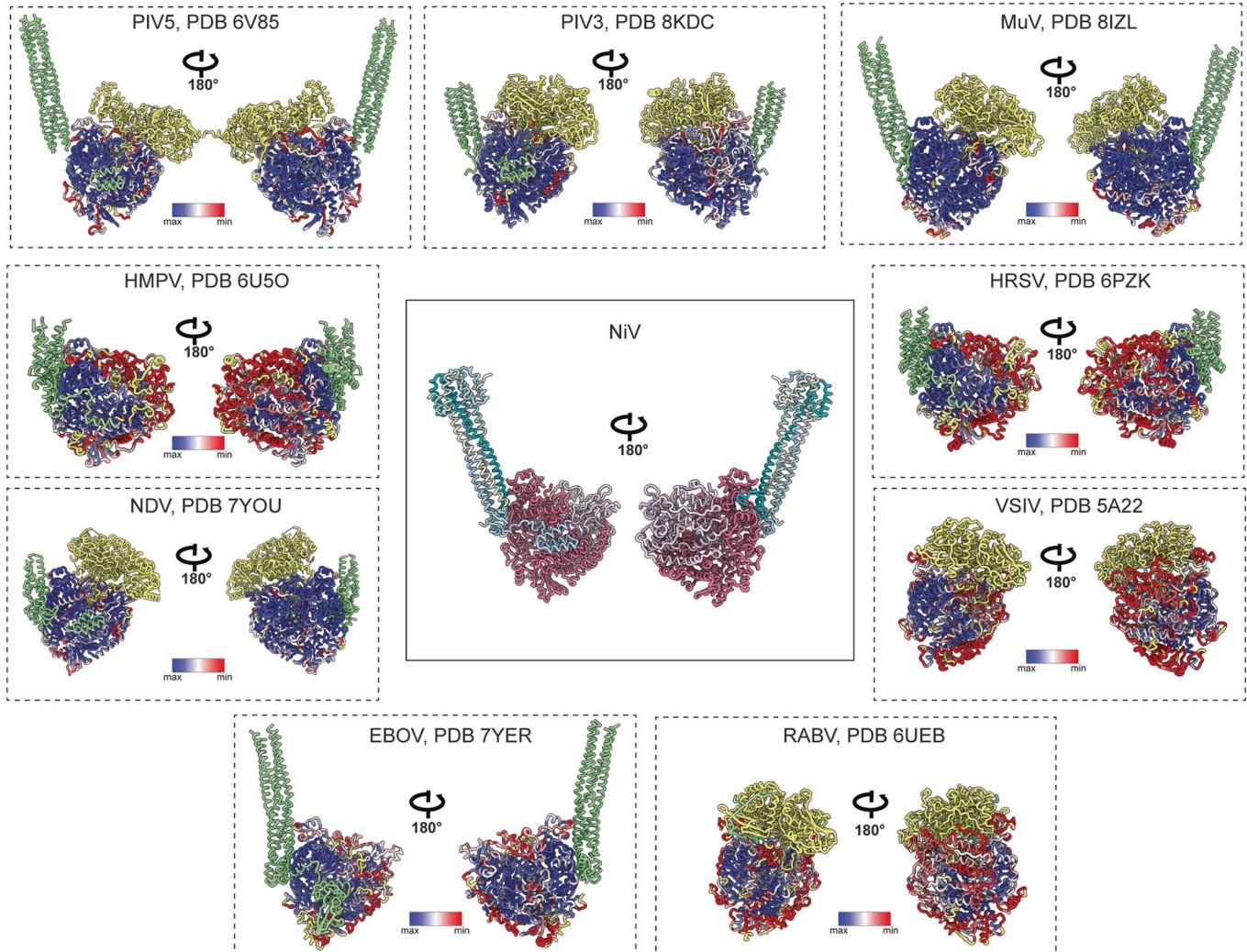

**Figure EV3. Comparisons of the whole L-P complex with homologous structures.**

The L-protein structure of NiV was aligned with homologous structures, including PIV5 (PDB 6V85, RMSD 1.223 Å), PIV3 (PDB 8KDC, RMSD 1.113 Å), MuV (PDB 8IZL, RMSD 1.182 Å), HMPV (PDB 6U5O, RMSD 1.338 Å), NDV (PDB 7YOU, RMSD 1.134 Å.), EBOV (PDB 7YER, RMSD 1.349 Å), HRSV (PDB 6PZK, RMSD 1.372 Å), VSIV (PDB 5A22, RMSD 1.297 Å), and RABV (PDB 6UEB, RMSD 1.252 Å) The structures were superimposed onto the NiV L-protein, and the RMSD values were calculated in ChimeraX. The sequence alignments of the L-protein structures were colour-coded based on residue conservation, with blue representing highly conserved residues and red indicating the least conserved. The residues present in the homologous L-protein structures but absent in NiV L-protein structure were coloured in yellow. For simplicity, the P-proteins of the homologous structures were shown in pale green.

## A  RdRp domains

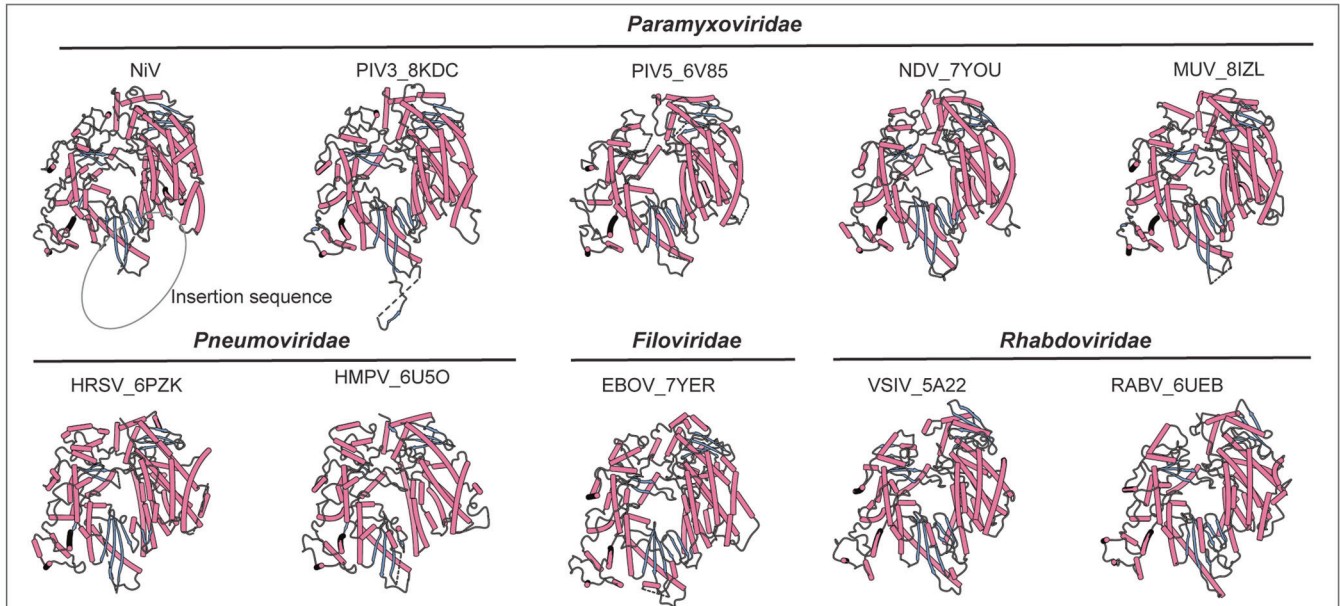

## B  PRNTase domains

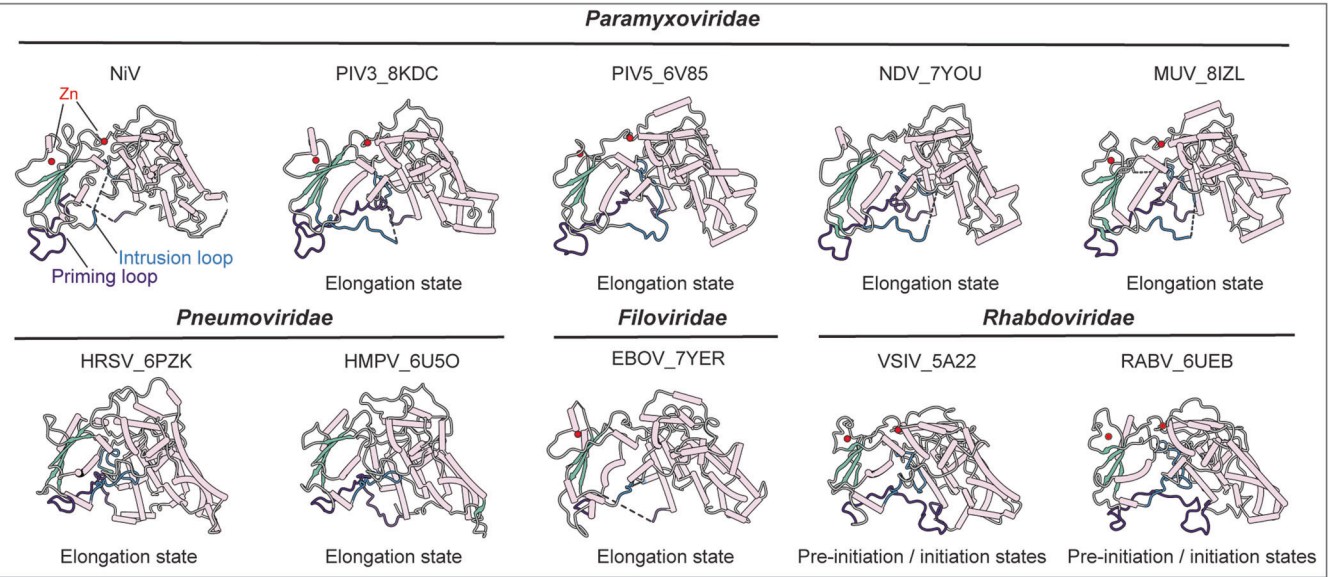

**Figure EV4.   Structural comparisons of RdRp- and PRNTase-domains of NiV L-protein with those of nsNSVs.**

(A) The RdRp-domains are shown with helices coloured in pale purple red and β-sheets in blue. The location of the unmodelled palm-insertion sequence is shown by a dashed circle. (B) The PRNTase-domains are represented using secondary structure colourings: helices are shown in pale pink, β-strands are in green. Zinc ions bound to the PRNTase-domains are coloured in red, while the priming loop is represented in dark purple and the intrusion loop in marine blue.

