## [Peer Review File · The EMBO Journal]

Structure of the Nipah virus polymerase complex

Jonathan Grimes, Esra Balıkçı, Franziska Guenl, Loïc Carrique, Jeremy Keown, and Ervin Fodor

Corresponding author: Jonathan Grimes (jonathan@strubi.ox.ac.uk)

Review Timeline:

Submission Date:	13th Oct 24
Editorial Decision:	4th Nov 24
Revision Received:	10th Nov 24
Accepted:	12th Nov 24

Editor: Ieva Gailite

Transaction Report:

(Note: Please note that the manuscript was previously reviewed at another journal and the reports were taken into account in the decision making process at The EMBO Journal. With the exception of the correction of typographical or spelling errors that could be a source of ambiguity, letters and reports are not edited. Depending on transfer agreements, referee reports obtained elsewhere may or may not be included in this compilation. Referee reports are anonymous unless the Referee chooses to sign their reports.)

Response to Reviewers Comments:

Reviewer #1 (Remarks to the Author):

The manuscript by Balikçi et al., presents the cryo-EM structure of the apo-Nipah virus polymerase complex comprising the viral L protein and a tetramer of the viral P protein, at an overall 2.5 Å resolution. As with many other L-P complexes, the connecting domain (CD), MTase and CTD domains are missing, presumably due to flexibility. To compensate, the authors determine by X-ray crystallography the structure of the CD domain at 1.85 Å resolution.

Although the Nipah L-P complex has not been previously described it turns out that it is very similar to that of related viruses (it would be helpful to have overall RMSDs in Supp. Fig. 3). A difference is in the interactions of some of the P protein monomers, notably P1, in which an extended C-terminal region (including the XD region) is observed. Also, Nipah L has an unusually long insertion in the palm, which the authors show, by deletion, is required for L-P activity in a cell-based minireplicon assay.

The manuscript describes generic aspects of the polymerase (e.g. the polymerase module has conserved motifs A-G, it has a NTP entry channel and RNA entry and exit channels) and uses AF2/AF3 to model the complete L protein and the template and product RNA, with small molecule cofactors (GTP, Mg etc). This allows speculation about how the exiting product contacts motifs involved in capping and suggests that the CD domain aids in positioning the GTP involved in capping.

The main problem with this paper is that there is an excessive amount of mechanistic speculation in both the results and the discussion based on the AF3 composite model with RNA, with absolutely no experimental validation. This, despite the fact that the authors demonstrate that they can study the activity of mutants based on in vitro assays or the minireplicon system. Similarly, there is no investigation of the functional importance of the observed L-P interactions that are structurally described in detail (e.g. the helices around the NTP entry channel). One even has the impression that much of this could have been written based on AF without having the experimental structure at all. This does not really advance understanding of how these dynamic machines perform RNA synthesis. Active, functional structures are required.

We thank the reviewer for their comments. We have included new results to validate the AF3 composite model. In particular, we generated mutations at amino acids proposed to play a role in capping and tested the resulting mutant L proteins both in vitro and in a minireplicon assay. As expected, based on the proposed model, mutations affected minireplicon activity that is dependent on capping, while the same mutations had no effect in the in vitro assays that measure the ability of L-P to generate RNA. These results are included as panels e,f in Figure 6. We have also addressed the functional importance of the observed L-P interactions by performing mutagenesis of amino acid residues at the L-P interface. As expected, mutations affected the L-P interaction as measured using a split-luciferase assay and the same mutations also affected polymerase activity in a minireplicon assay. These data are of Supplementary Figure 7.

We have also included overall RMDs in Supplementary Figure 3 as suggested by the Reviewer.

Specific points.

(1) The activity assays and their presentation in Supp. Fig 1c-e needs improving (compared to those in the cited publication [7], they are very messy). In particular, (a) in the Methods it says a 20-mer DNA was used as a marker, but the marker band is not visible on the gel. It would be better to use at least two RNA markers. (b) Supp. Fig. 1d should be on the same scale as 1c and 1e., (c) it is not clearly stated whether a primer was used for 1d and 1e. (d) When just G is added in 1c and 1d, there should be no products, however in lane 2 of 1c there are (but curiously not in lane 1 of 1d, which is the same reaction). The authors should explain and annotate spurious bands.

We have repeated the activity assays and improved the presentation. Now a radiolabelled RNA 12mer has been used as a marker. Degradation products of this marker indicate mobility of 11 nt long and shorter RNA products. We now present all the experiments run with the wild type leader sequence in one panel (Suppl. Fig. 1 c), including the controls leaving out the L-P complex, the template, additional nucleotides, the primer, as well as using the RdRp active site mutant (L-D832A). The experiment run with the mutant (5C) template to assess the impact of Remdesivir incorporation is now being presented in Suppl. Fig. 1d. In both Suppl. Fig 1c and d, lane 3 or 2, respectively, we see a thick band at approximately 5 nt. We assume that this product stems from the addition of one GTP to the primer opposite of the first UTP.

(2) In the CD domain crystal structure, how were the magnesium ions identified as such?

We thank the reviewer for the insightful comment regarding the identification of the bound metal ion in our structure. To ensure accurate assignment of the Mg^{2+} in our structure, we performed several analyses and refinements. Magnesium ions normally have octahedral coordination geometry and the ligand distances and angles are consistent with typical Mg^{2+} coordination. We further explored the possibility of Mn^{2+} being bound at the sites by fitting and refining Mn^{2+} ions in our structure. However, the refinement led to significantly higher B-factors compared to the coordinating protein sidechains, showing that the crystallographic refinement inflates the atomic B-factor to compensate for the higher scattering factors of Mn compared to Mg. This strongly suggests that Mg^{2+} , rather than Mn^{2+} , is the preferred ion at this site in our structure. We have revised our text to include these explanations in the result section “Structural details of the polymerase complex” for clarity.

(3) Lines 207-208. The authors state that the new composite model predicts with confidence the pathway of the template RNA... However in Supp. Fig 7, the RNA is red to orange indicating very low to low confidence.

We acknowledge this point and have removed “with confidence” from the text.

(4) The sliding model remains speculative and is not based on significant additional experimental data in the paper, therefore should be moved to a supplementary figure. We agree that the sliding model, while an interesting hypothesis, is speculative without further experimental data. We moved it to a supplementary figure¹² as recommended to maintain focus on the primary findings in the main text.

Reviewer #2 (Remarks to the Author):

This is an important paper describing an experimental structure of the Nipah virus polymerase complex.

The work needs to be published rapidly because it has high impact as (correctly stated by the authors in the abstract) Nipah virus is a recurring severe respiratory virus with very high lethality. In view of the necessary preparedness (eg for screening antivirals) this well-executed structural study using SP cryo-EM gives key information that can be exploited for instance like Fearn and coll. did for RSV pol complex (see for instance Structural and mechanistic insights into the inhibition of respiratory syncytial virus polymerase by a non-nucleoside inhibitor.

Yu X, et al. Commun Biol. 2023 Oct 21;6(1):1074. doi: 10.1038/s42003-023-05451-4.)

Conserved allosteric inhibitory site on the respiratory syncytial virus and human metapneumovirus RNA-dependent RNA polymerases.

Kleiner VA, O Fischmann T, Howe JA, Beshore DC, Eddins MJ, Hou Y, Mayhood T, Klein D, Nahas DD, Lucas BJ, Xi H, Murray E, Ma DY, Getty K, Fearn R. Commun Biol. 2023 Jun 19;6(1):649. doi: 10.1038/s42003-023-04990-0.

Overall, the study is remarkable and very well presented.

I would have liked to see a bit of inhibitor modelling or at least discussion (after all we now have alphafold3 server) based on the current RNA model (Fig. 6) and publications like those above for a NNI/allosteric inhibitors and also for some nucleoside inhibitors.

We thank reviewer for the positive feedback and for emphasizing the importance of our study. We appreciate the reviewer's suggestion for including inhibitor modelling. While we believe this would be a valuable addition, given the scope of our current study, which primarily focuses on the structural characterization of the Nipah polymerase complex, we have decided to defer detailed inhibitor modelling to future investigations. As more structural data on potential inhibitors becomes available, we anticipate that such analysis will be instrumental in guiding the development of antiviral strategies. We hope that the structural insights presented in this study serve as a foundation for future research aimed at targeting Nipah virus polymerase.

Reviewer #3 (Remarks to the Author):

The manuscript by Esra Balikci et al. describes a structural biology study on Nipah virus polymerase complex. The authors performed cryoEM and X-ray crystallography experiments, which were supplemented by AlphaFold 3 modeling. The manuscript is easy to read and

understand, but I think that it can be improved significantly by adding just a few corrections, which are especially necessary in the computational structure prediction part.

The abstract of the manuscript mentions only experimental structure determination, but some of the conclusions are based on the computational structure modeling. Therefore I would suggest to also mention the structure modeling in the abstract.

We thank reviewer. We appreciate this suggestion to clarify the role of AlphaFold modelling in our study. We revised the abstract to explicitly state that our conclusions are based on a combination of experimental structure determination and computational modelling.

A number of structures of homologous protein complexes of viral polymerases are already available in the Protein Data Bank. Some of them contain protein-protein and protein RNA interactions homologous to the investigated L-P complex (I found them easily using the PPI3D server). The authors mention homologous structures in several places of the manuscript, and the similarity of the structures of individual domains is briefly discussed in Discussion section and supplementary figures. However, I did not notice the comparison of the whole complex structure with what was known from the previous studies of these homologs. This would better place the described results into the context.

We thank reviewer for the valuable feedback. We agree that comparing the whole L-P complex with homologous structures would provide important context. We have performed these additional comparisons and included the results in Supplementary Fig. 3 in the revised manuscript.

When searching for the information on the topic of the manuscript, I found a highly similar preprint from a different research group (<https://www.biorxiv.org/content/10.1101/2024.05.29.596445v1.full>). It was not referenced in the current manuscript, and it would be interesting to see if the conclusions are similar.

We thank reviewer for pointing out the related preprint. We have included a comparison of their data in the Discussion

The authors used AlphaFold 3 to predict interactions with RNA and to predict the complex structure with all domains, including the ones not visible in the experimental structure. I have a very serious remark on the structure modeling part. Before the description of the AlphaFold 3 models, and before making any conclusions based on these models, the authors should provide the analysis at least their accuracy self-estimation data, such as pLDDT, pTM scores and predicted aligned error (PAE) plots. Especially the latter plots would be highly useful to understand if the domain orientation is predicted confidently or not. From what I see now in Supplementary Fig. 7, the core of the protein is predicted well (blue part, pLDDT ~0.9), but experimental structure of this part is probably available. The terminal domains, which are not visible in the experimental structure, are predicted worse (pLDDT 0.7-0.8), and RNA is predicted with only low confidence. The PAE plot is not given (which should be

mandatory for multimeric complexes and multidomain proteins), thus it is not clear whether AlphaFold itself trusts the prediction of the domain orientation. This creates doubts regarding all the further conclusions derived from this structure model.

We understand the importance of including accuracy self-estimation data to support our conclusions. In response to your comments, we have incorporated the pLDDT, pTM scores, and PAE plots to provide a clearer assessment of model accuracy and domain orientation. These additional plots are included in Supplementary Figure 8 in the revised manuscript to address the concerns and enhance the analysis

In addition to accuracy self-estimates from the neural network model, AlphaFold-independent structure analysis methods, even such as old-school statistical potentials ProSA, QMEAN or VoronMQA, might be also helpful when the model is ambiguous.

In my opinion, even a structure model having lower confidence self-estimates can be useful for further investigation, but in this case all the data should be handled with care. A more thorough analysis of the homologous structures could help in this situation. I am not an expert in viral polymerases, but there are structures having structured C-terminal CD and MTase domains (I found PDB 8KDC, 8IZL after a quick search). There are structures with RNA (authors mention EBOV polymerase structure, there is also hPIV3 structure in PDB entry 8KDC). As a result, the authors could probably expand and clarify the section “A model for RNA binding and processing” and the Discussion part by taking the data on homologous structures into account, and by showing how their AlphaFold 3 model agrees with previous experiments.

To address the accuracy of our model, as suggested, we conducted an analysis using the HRSV RNA-bound structure (PDB 8SNX) and showed the predicted model agrees with the experimental structure. In addition we also ran predictions for PIV L. The PIV3 L-P complex structure (PDB 8KDC), which has resolved CD, MTase, and CTD the domains and this comparison allowed us to demonstrate how the predicted positions of these domains align well with experimentally resolved structures. Unfortunately, due to restrictions imposed by the AlphaFold server, certain viral pathogen sequences, including the EBoV RNA-bound structure, are filtered out, limiting our ability to include this structure (PDB 8JSL) in the

prediction. These data are shown in the Figure below.

AF3 model predictions of structurally determined RNA-bound L-P complexes and L-P complexes with modelled CD, MTase, and CTD domains. For the prediction of the RNA-bound HRSV structure, we used the sequences and ions based on PDB ID: 8SNX. The full-length sequence of HRSV L (UniProt ID: P28887) and four copies of HRSV P (residues 120–241) (UniProt ID: P03421) were used, along with the RNA sequence 5'-UUUUUCGCGU-3'. For the PIV3 structure prediction based on PDB ID: 8KDC, we used the full-length PIV3 L sequence (UniProt ID: O89238) and four copies of the PIV3 P sequence (residues 430–603) (UniProt ID: O89234). Additionally, two Zn ions and one Mg ion were included in the prediction. **a.** The structure prediction for the PIV3 L-P complex is shown, coloured by pLDDT confidence scores. The corresponding PAE plot is displayed at the bottom of the panel, with ipTM = 0.61 and pTM = 0.72. **b.** The RNA-bound structure of the HRSV complex

is shown, with colouring based on pLDDT scores. The related PAE plot is presented at the bottom of the panel, with ipTM = 0.58 and pTM = 0.69. **c.** Superimposition of the predicted RNA-bound HRSV L-P complex and the experimentally validated structure (PDB 8SNX), with RMSD of 0.581 Å. A close-up view of the RNA-binding site shows that the predicted RNA and the experimental structure exhibit highly similar binding modes. **d.** Superimposition of the predicted PIV3 L-P complex and the experimentally resolved structure (PDB 8KDC), demonstrating close alignment of the predicted domain positions with the solved structure. The RMSD for the superimposition is 1.083 Å.

The AlphaFold 3 model itself is not made available anywhere. I do not know what are the current requirements and traditions regarding computational models. It's publication could be also hindered by the licensing of AlphaFold Server. If the structure model is not made available, I would suggest the authors to at least provide not only protein sequences, but also the RNA sequences and the number of Mg ions that were used for modeling, as well as the seed used by AlphaFold 3. In such way the interested readers could generate a similar structure model themselves using the AlphaFold Server.

We appreciate the reviewer's suggestion regarding the AlphaFold 3 model. Although the model itself is not publicly available, we have included the exact protein sequences, RNA sequences, the number of ions used, as well as the nucleotides and their copy numbers in the methods section of the revised manuscript. This information should enable interested readers to generate similar structure models using the AlphaFold Server.

It was not clear for me which experimental or modeling data form the basis for the mechanism depicted in Fig. 7.

We agree that the sliding model in Fig. 7, though an intriguing hypothesis, remains speculative without additional experimental data. To address this, we moved it to a supplementary figure 12.

Other smaller remarks:

- In multiple places through the text, the proteins and domains are referred to using just one letter name or two letters. It would be easier to read, if "P protein", "L protein", "OD domain", "CD domain", etc. would be written.

We agree with the reviewer's suggestion and have revised the manuscript to use the recommended terms, such as "P-protein," "L-protein," "OD-domain," and "CD-domain," to enhance readability.

- In the introduction and in some further places, references to Fig. 1 where the domain organization of the investigated proteins is shown would be highly beneficial to the readers.

We thank reviewer for the helpful suggestion. We agree that referencing Fig. 1 in the introduction and other relevant sections would enhance the clarity and usefulness for readers. We will include references to Fig. 1 where the domain organization of the investigated proteins is discussed.

- A typo “Supplementary” is in line 117.

We thank the reviewer for noting the typo "Supplementary" on line 117. We have corrected it in the revised manuscript.

- A reference to the EBOV RNA polymerase structure could be given in lines 150-151. We have added a reference to the EBOV RNA polymerase structure in lines 150-151 of the revised manuscript.

- The colors in Fig. 1 could provide more contrast to distinguish different domains/proteins/chains.

We thank reviewer for the suggestion. We appreciate the importance of clear differentiation in color contrast. We have carefully balanced contrast and coherence in our current color scheme to support data interpretation. While we understand the value of adjusting colors for better contrast, making changes could affect the consistency of our figures throughout the manuscript and add complexity due to the multiple colors already used.

Dear Jon,

Thank you for submitting your revised manuscript to The EMBO Journal. Your manuscript has now been seen by two of the original reviewers, who now both broadly support publication of the manuscript here. I will therefore be happy to accept the manuscript for publication in The EMBO Journal after its minor textual revision long the lines requested by reviewer #3, especially regarding discussion of similarities and differences to other recently published structures of the Nipah virus polymerase complex. Furthermore, please reformat the manuscript along the guidelines included in the attached document.

Please feel free to contact me if you have any further questions regarding this final revision. You can use the link below to upload the revised files.

Thank you for the opportunity to consider your work for publication, and I look forward to receiving your revised manuscript.

With best regards,

leva

leva Gailite, PhD
Senior Scientific Editor
The EMBO Journal
Meyerhofstrasse 1
D-69117 Heidelberg
Tel: +4962218891309
i.gailite@embojournal.org

We realize that it is difficult to revise to a specific deadline. In the interest of protecting the conceptual advance provided by the work, we recommend a revision within 3 months (2nd Feb 2025). Please discuss the revision progress ahead of this time with the editor if you require more time to complete the revisions.

Referee #2:

This is a thoroughly revised version of a manuscript reporting the structure of NiV polymerase complex. In particular, data shown in Fig. 6. and Fig S1 and S9 bring important functional validation of the cryo-EM structural work presented. As the reviewer's concerns have been addressed, I would support publication.

Referee #3:

The authors answered all of my questions of the first review round, but one answer seems a bit strange to me. I was asking for comparison of AlphaFold 3 (AF3) models with available experimental structures of homologous proteins. As I understood from the reply to reviewers, the authors did a remodeling of some known structures with AF3, and then they compared the models with the structures. In my opinion, it would be more reasonable to compare their model of the Nipah polymerase which is investigated in the current study with the previously released experimental structures of other viral polymerases. Isn't it so?

Also, this expanded study with additional AF3 modelings had a good aim to confirm the accuracy of performed structure modeling, but it is described only in the reply to reviewers. After some clarifications regarding the issue mentioned above, maybe a place for this text could be found, for example, in the supplementary data of the manuscript? As it is now in the manuscript, the structure model seems to be trustful without further validations, and I think that this is not a good message from an experimental structural biology paper. All computational models should be handled with care.

The authors also cited two similar studies ("additional structures of the NiV L-P complex were also made available online [36, 37]"), but did not go into details how their results differ from these concurrently published works.

Minor comments:

* Fig. 1a could benefit from annotations where is which protein.

* A longer description about the evaluation of AF3 model quality in the methods part could be useful for the readers (and it could also point to the validation study that is mentioned above).

Response to Reviewers Comments:

The authors also cited two similar studies ("additional structures of the NiV L-P complex were also made available online [36, 37]"), but did not go into details how their results differ from these concurrently published works.

This text has been updated and includes a new paragraph (bottom of page 9, line 283))

"During the submission of this paper, additional structures of the NiV L-P complex were also made available online [36, 37], revealing a conformation similar to our own NiV L-P complex structure, including the conformations of the RdRp- and PRNTase-domains and the binding of P-protein to the L-protein. However, the NiV L-P structure presented here provides distinct insights on structure and function within the broader context of the complete polymerase, as discussed in detail below."

Minor comments:

* Fig. 1a could benefit from annotations where is which protein.

This has been done

* A longer description about the evaluation of AF3 model quality in the methods part could be useful for the readers (and it could also point to the validation study that is mentioned above).

We have now included the figure in response to the reviewer as another figure in the Appendix. S7.

Dear Jon,

Thank you for addressing the final formatting issues. I am now happy to inform you that your manuscript has been accepted for publication in the EMBO Journal.

Before we forward your manuscript for typesetting, I would like to propose some minor edits in the manuscript synopsis and abstract. I have also written a short blurb that will accompany the title of your study in our online table of contents. Please take a look at the text in the attached file and let me know if any corrections are needed.

Your manuscript will be processed for publication by EMBO Press. It will be copy edited and you will receive page proofs prior to publication. Please note that you will be contacted by Springer Nature Author Services to complete licensing and payment information. We have requested rapid processing of your manuscript, so you should hear from our publishers soon.

If you have any questions, please do not hesitate to contact the Editorial Office. Thank you for your contribution to The EMBO Journal and congratulations on a nice study!

With best wishes,

Ieva

Ieva Gailite, PhD
Senior Scientific Editor
The EMBO Journal
Meyerohofstrasse 1
D-69117 Heidelberg
Tel: +4962218891309
i.gailite@embojournal.org
